# A versatile *Plasmodium falciparum* reporter line expressing NanoLuc enables highly sensitive multi-stage drug assays

Yukiko Miyazaki [1,2,9]✉, Martijn W. Vos[3], Fiona J. A. Geurten[2], Pierre Bigeard[4], Hans Kroeze[2], Shohei Yoshioka[5], Mitsuhiro Arisawa[5], Daniel Ken Inaoka[1,6,7], Valerie Soulard[4], Koen J. Dechering [3], Blandine Franke-Fayard[2] & Shinya Miyazaki [2,8]✉

Transgenic luciferase-expressing *Plasmodium falciparum* parasites have been widely used for the evaluation of anti-malarial compounds. Here, to screen for anti-malarial drugs effective against multiple stages of the parasite, we generate a *P. falciparum* reporter parasite that constitutively expresses NanoLuciferase (NanoLuc) throughout its whole life cycle. The NanoLuc-expressing *P. falciparum* reporter parasite shows a quantitative NanoLuc signal in the asexual blood, gametocyte, mosquito, and liver stages. We also establish assay systems to evaluate the anti-malarial activity of compounds at the asexual blood, gametocyte, and liver stages, and then determine the 50% inhibitory concentration ($IC_{50}$) value of several anti-malarial compounds. Through the development of this robust high-throughput screening system, we identify an anti-malarial compound that kills the asexual blood stage parasites. Our study highlights the utility of the NanoLuc reporter line, which may advance anti-malarial drug development through the improved screening of compounds targeting the human malarial parasite at multiple stages.

[1] Department of Molecular Infection Dynamics, Institute of Tropical Medicine (NEKKEN), Nagasaki University, 852-8523 Nagasaki, Japan. [2] Department of Parasitology, Leiden University Medical Center, 2333 ZA Leiden, The Netherlands. [3] TropIQ Health Sciences, Transistorweg 5, 6534 AT Nijmegen, The Netherlands. [4] Sorbonne Université, Inserm, CNRS, Centre d'Immunologie et des Maladies Infectieuses, CIMI-Paris, F-75013 Paris, France. [5] Graduate School of Pharmaceutical Sciences, Osaka University, 565-0871 Osaka, Japan. [6] School of Tropical Medicine and Global Health, Nagasaki University, Nagasaki 852-8523, Japan. [7] Department of Biomedical Chemistry, Graduate School of Medicine, The University of Tokyo, Tokyo 113-0033, Japan. [8] Department of Cellular Architecture Studies, Institute of Tropical Medicine (NEKKEN), Nagasaki University, 852-8523 Nagasaki, Japan. [9] Present address: Department of Protozoology, Institute of Tropical Medicine (NEKKEN), Nagasaki University, Nagasaki, Japan. ✉email: y.miyazaki@nagasaki-u.ac.jp; smiyazaki@nagasaki-u.ac.jp

*P*lasmodium falciparum parasites cause malaria, a severe infectious disease, which represents a major public health issue, especially in sub-Saharan Africa[1]. The life cycle of *P. falciparum* in human hosts consists of pre-erythrocytic stages in liver cells, followed by asexual blood and sexual stages in red blood cells (RBCs)[2]. After an *Anopheles* mosquito bite, *P. falciparum* parasites (sporozoites) enter the human liver and invade hepatocytes wherein they multiply. Parasites in the hepatocytes are eventually released into the bloodstream, and subsequently invade RBCs for further proliferation. RBCs subsequently rupture and release replicated parasites (asexual blood stage). These asexual blood parasites repeatedly invade, replicate, and egress in the bloodstream, with a certain proportion developing a sexual form (gametocyte) which is transmissible into *Anopheles* mosquitoes[3,4].

Since parasites are capable of causing malaria symptoms only at the asexual blood stage, anti-malarial drug development has focused on targeting asexual *P. falciparum* parasites, with existing anti-malarial drugs, such as artemisinin, being primarily effective. However, to control malaria, the prevention of *P. falciparum* infection and transmission is also essential. Therefore, it is imperative to develop novel anti-malarial drugs targeting the parasite at multiple stages, namely liver-stage parasites, gametocytes, and asexual parasites[5–7].

To identify compounds preventing malarial transmission, robust, easy-to-use, and simple high-throughput screening (HTS) techniques using gametocytes need to be established. To date, a variety of drug assay systems have been developed for targeting gametocytes based on ATP consumption, viability assays using Mitotracker or alamarBlue, and the evaluation of gametogenesis[6,8–11]. However, these methods are not user-friendly owing to the technical complexities associated with the purification of synchronised gametocytes and the need for expensive high-content imaging devices[8,9]. Furthermore, to develop drugs to prevent *P. falciparum* infection, the evaluation of compounds against liver-stage *P. falciparum* was performed. However, this approach is still technically challenging owing to the difficulty in handling primary human hepatocytes, requirement of costly devices, and extremely low infectivity of *P. falciparum* sporozoites[12].

Transgenic human or rodent malaria reporter parasites have been extensively used in parasite biology, including those for the development of interventions, such as anti-malarial drugs and vaccines[13]. Various transgenic *P. falciparum* lines have been generated expressing a fluorescent protein and/or luciferase in the asexual blood stage, gametocyte stage, oocyst, sporozoites, and liver stages[14–22]. Fluorescent reporter parasites have certain advantages in detection by microscopy or sorting by flow cytometry during their specific life cycle[23–25]. Luminescence from expressed luciferase has been used as a readout for drug assays against *P. falciparum* at multiple life cycle stages[17,26–31]. Although HTS for gametocytocidal drugs, which utilises luciferase-expressing *P. falciparum*, was recently achieved by a relatively robust and simple assay method, the reporter line has not been optimised for assays targeting asexual blood- and liver-stage parasites because luciferase expression is confined to the gametocyte stage[32].

To establish multi-stage *P. falciparum* drug assays, we generated marker-free transgenic *P. falciparum* NF54 reporter parasites expressing green fluorescent protein (GFP) and NanoLuciferase (NanoLuc) under a constitutive promoter. NanoLuc is a suitable reporter protein for high throughput screening, which has a higher luminescence intensity (80- to 240-fold) and structural stability than firefly luciferase[33]. We previously reported that the 40S ribosomal protein promoter is constitutively active throughout the life cycle of *P. falciparum*[16]. The constitutive expression of NanoLuc under this promoter permits highly sensitive drug assays at the asexual blood, gametocyte, and liver stages of *P. falciparum*. Furthermore, we employed this transgenic parasite to identify a novel compound that killed the parasite at multiple stages.

## Results

### Generation of transgenic *P. falciparum* reporter lines expressing both fluorescent protein and NanoLuc.

To generate a versatile reporter line that can be used for multi-stage drug assays, we introduced an expression cassette containing GFP-T2A-NanoLuc or mCherry-T2A-NanoLuc into the *p47* locus of the *P. falciparum* NF54 strain using CRISPR/Cas9 (Fig. 1a, Supplementary Fig. 1a, plasmid maps are shown in Supplementary Fig. 2a)[16]. To achieve constitutive expression throughout the life cycle of *P. falciparum*, we used the 40 S ribosomal protein S30 promoter (PF3D7_0219200), which is highly active at all life cycle stages, as previously determined[16]. We used the T2A skip peptide coding sequence to express two different reporter proteins from a single expression cassette[34]. Transfected *P. falciparum* parasites were subjected to double-positive selection using WR99210 and blasticidin along with subsequent negative selection using 5-fluorocytosine. PCR analysis of genomic DNA from the transgenic parasites confirmed the appropriate integration of the desired GFP-NanoLuc expression cassettes into the *p47* locus of *P. falciparum* NF54 (which we termed as "Exp245 clone1", Fig. 1b). We also confirmed the integration of the donor plasmid containing the mCherry-NanoLuc expression cassette (which we termed as "Exp221 uncloned", Supplementary Fig. 1b). In vitro growth of the GFP-NanoLuc line at the asexual blood stage was comparable to the growth at the asexual blood stage of the parent NF54 WT strain (Fig. 1c), producing normal numbers of stages III–V gametocytes (Table 1). The lack of apparent developmental defects caused by targeting the *p47* locus in NF54 strain is consistent with previous findings[14,16,18].

We then validated efficient skipping of the T2A coding sequence in the expression cassette. Western blotting analysis revealed a major band at around 25–37 kDa, which is consistent with the expected size of the GFP (Fig. 1d). We also detected a minor product corresponding to a non-skipped GFP-NanoLuc fusion protein of ~50 kDa (Fig. 1d). This result confirmed the expression of the two different reporter proteins, GFP and NanoLuc.

### Analysis of reporter expression and validation of drug assay using the GFP-NanoLuc reporter line in the asexual blood stage.

We investigated fluorescent protein expression at the asexual blood stage using fluorescence microscopy. Strong fluorescent signals of GFP or mCherry were observed in the ring, trophozoite, and schizont asexual blood stages (Fig. 2a, Supplementary Fig. 1c). Quantitative analysis of the luminescence derived from NanoLuc exhibited a clear correlation with the number of *P. falciparum*-infected RBCs during the asexual blood stage (Fig. 2b). Subsequently, we successfully obtained a clonal line of the GFP-NanoLuc reporter line, but not the mCherry-NanoLuc line. For downstream analysis, we mainly used the GFP-NanoLuc reporter line owing to the clonality of an isolated clonal line (cl.1) and the utility of GFP for sorting specific life cycles, as exemplified by several studies wherein different *Plasmodium* species expressing GFP were employed[25,35,36].

Next, we evaluated the utility of this GFP-NanoLuc reporter line for an anti-malarial drug assay in the asexual blood stage. The synchronised ring-stage GFP-NanoLuc reporter parasites were seeded in 384-well plates with various anti-malarial compounds (atovaquone, dihydroartemisinin (DHA), pyrimethamine,

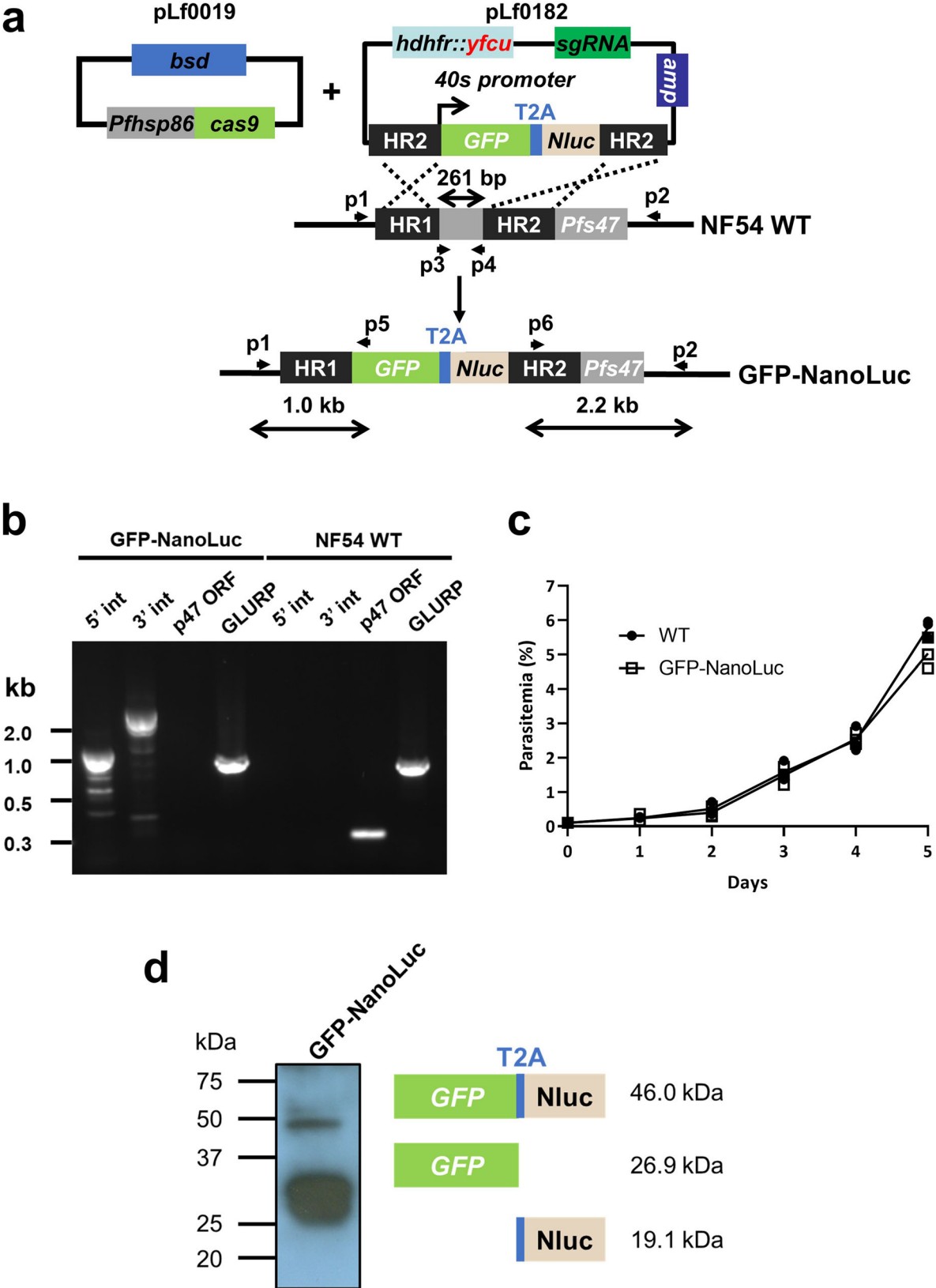

puromycin, DSM265, epoxomicin, WR99210, and blasticidin). The NanoLuc luminescence level was measured after 72 h of incubation (Fig. 2c). IC$_{50}$ values of the anti-malarial compounds were also determined and were comparable with those reported previously[15,28,37–39], thus validating the GFP-NanoLuc reporter line for drug assays in the asexual blood stage (Fig. 2d, Table 2).

Importantly, the GFP-NanoLuc reporter line was sensitive to WR99210 and blasticidin, thereby confirming that there was no retention of the plasmids used for genetic modification (Fig. 2d, Table 2). The statistical parameters for assay validation, such as signal-to-background ratio (S/B), signal-to-noise ratio (S/N), maximum coefficient of variation (CV$_{max}$), minimum of CV

**Fig. 1 Generation of *P. falciparum* NF54 reporter lines expressing GFP and NanoLuc under the control of the 40S promoter. a** CRISPR plasmids used for generation of the GFP-NanoLuc reporter line. The Cas9 expression plasmid (pLf0019) and donor DNA/gRNA plasmid (pL0182) constructs were used to introduce the GFP-NanoLuc expression cassette with T2A into the *P. falciparum* NF54 *p47* gene locus. *p47* homology regions (HR1, HR2) used to introduce the donor DNA, location of primers (p) and the expected PCR products (in black) are indicated. Primer sequences are shown in Supplementary Table 1. WT, wild-type; bsd, blasticidin selectable marker (SM); hdhfr::yfcu, WR99210-5-FC SM in donor plasmid; Nluc, NanoLuciferase. **b** Genotyping analysis confirmed the appropriate integration of the donor plasmids into the genome of the clonal GFP-NanoLuc reporter line (Exp245 clone1, 5-Int; primers p1/p5 for 1,087 bp, 3-Int; primers p6/p2; 2,188 bp). Primer positions and the expected DNA sizes are shown in (**a**). The PCR product of the WT *p47* gene amplified by p3/p4 primers was detected only in the WT genomic DNA sample. Amplification of the *glurp* gene was used as a positive control for the PCR reaction. **c** Growth phenotype of asexual blood stages of the GFP-NanoLuc reporter line. Parasitaemia of NF54 WT and GFP-NanoLuc reporter line is shown during a 5-day culture period in the static culture system. Cultures were initiated with a parasitaemia of 0.1%. Each parasitemia of three independent cultures is shown. **d** Western blot analysis of the fusion protein GFP-NanoLuc showing efficient T2A-peptide-mediated cleavage of the protein. The protein was extracted from mixed asexual blood stage parasites. Separated proteins were stained with a rabbit polyclonal anti-GFP antibody (Abcam; ab290).

| Table 1 Gametocyte, infection ratio, oocyst and sporozoite production of the GFP-NanoLuc reporter line. | | | | | |
|---|---|---|---|---|---|
| Lines | Stage III Gametocytemia[a] | Stage V Gametocytemia[a] | Infection ratio | No. of oocyst[b] | No. of sporozoites (×10³)[c] |
|  | Average (SD) | Average (SD) | Average | Average (SD) | Average (SD) |
| NF54 WT | 1.0% (0.5%) (3 exp.) | 2.5% (0.9%) (6 exp.) | 86% (10 exp.) | 85 (60) (10 exp.) | 87,4 (36,7) (10 exp.) |
| GFP-NanoLuc Exp245 cl.1 | 1.7% (0.2 %) (4 exp.) | 2.0% (1.1%) (2 exp.) | 91% (5 exp.) | 69 (65) (5 exp.) | 91,6 (54,9) (5 exp.) |

exp, the number of biological replicates.
[a]Percentage of stage III or V gametocytes per 100 red blood cells in day 7 or 14 cultures, respectively.
[b]Mean number of oocysts per mosquito at day 7–12 after feeding (10–30 mosquitoes per experiment).
[c]Mean number of salivary gland sporozoites per mosquito at day 21 after feeding (20–30 mosquitoes per experiment).

($CV_{min}$), and Z'factor were $41.3 \pm 5.9$, $484 \pm 71$, $3.6 \pm 0.6\%$, $8.4 \pm 1.4\%$, and $0.88 \pm 0.02$, respectively (Table 3), thus highlighting assay robustness.

**Analysis of reporter expression and validation of the drug assay using the GFP-NanoLuc reporter line at the gametocyte stage.** To characterise the development and reporter expression during gametocytogenesis, we produced gametocytes of the clonal GFP-NanoLuc reporter line. The GFP-NanoLuc line produced a normal number of stages III–V gametocytes (Table 1). In addition, the stage III, IV, and V gametocytes of GFP/mCherry-NanoLuc reporter lines strongly expressed GFP or mCherry (Fig. 3a, Supplementary Fig. 1d), confirming constitutive expression under the 40S promoter.

To establish a gametocyticidal assay using this GFP-NanoLuc line, we first performed the same procedure as the drug assay for asexual parasites in a 96-well plate. However, we found that this method was not suitable for gametocyte-stage parasites, as the Z'factor of the assay was <0.5, which is caused by an extremely low S/B (Table 3, NanoLuc inhibitor (-), 72 h). This low S/B is due to the low signal of gametocytes in the negative control wells (DMSO), not increasing during 72 h of incubation, unlike that observed for asexual parasites. In addition, we speculate that the residual activity of NanoLuc proteins derived from killed gametocytes was still detected. To address these issues, the medium was replaced after 72 h of incubation to remove residual NanoLuc proteins, followed by additional incubation time for 72 h without any compounds[40]. Furthermore, the extracellular NanoLuc inhibitor (Promega) was added along with the substrates when luminescence was detected (Fig. 3c). These strategies dramatically improved gametocyticidal assay robustness (Table 3, 72 h + 72 h). The S/B rose to 21.2 and 14.5 (two independent experiments), and the Z'factor increased to 0.91 and 0.75. Gametocytes of the GFP-NanoLuc reporter line exhibited comparable sensitivity to epoxomicin, a positive control compound; the latter's sensitivity having been previously reported (Fig. 3d, Table 2)[41]. When the additional incubation time after

medium change was shortened to 0, 24, or 48 h, the S/B decreased, while Z'factors remained high ($0.80 \pm 0.08$, 0.95, and 0.81, respectively) (Table 3; 72 h + 0 h, 72 h + 24 h, 72 h + 48 h). In addition, the high assay quality was retained with a high Z'factor (0.89) when the medium change step was skipped (Table. 3, NanoLuc inhibitor (+) 72 h). This result indicates that the addition of extracellular NanoLuc inhibitor to assay reaction is critical for maintaining the assay quality and the medium change step is not essential. The $IC_{50}$ values of epoxomicin were similar under all assay conditions (Fig. 3d, Table 2). In contrast, atovaquone did not have any inhibitory effect on gametocytes at 1 μM, as previously reported (Fig. 3d, Table 2)[9]. These findings demonstrated that the GFP-NanoLuc line is applicable for *P. falciparum* gametocytocidal assays.

Furthermore, to examine whether this reporter line could be applied for HTS at the gametocyte stage, we used a gametocytocidal assay system in a 384-well plate. Since we found that the medium change step was not essential for high assay quality in a 96-well plate, this step was skipped, and luciferase assay with the extracellular NanoLuc inhibitor was performed after 72 h of incubation with the test drugs to simplify the HTS procedure (Supplementary Fig. 4a). Assay quality was retained with a high S/B ratio (8.4) and Z'factor (0.65) (Table 3). The inhibitory effects of epoxomicin and atovaquone were comparable with the results obtained when 96-well plates were used (Table 2), indicating that the 384-well format drug assay can be used for HTS.

**Analysis of GFP and NanoLuc expression using the GFP-NanoLuc reporter line at the mosquito stage.** To characterise reporter expression and parasite development in the mosquito host, we fed the gametocytes of GFP-NanoLuc reporter line to *Anopheles stephensi* mosquitoes and investigated midgut oocysts and salivary gland sporozoites. The GFP-NanoLuc reporter line infects *Anopheles* mosquitoes with an infection ratio comparable to that of NF54 WT and the feeding produced oocysts and sporozoites, whose numbers were comparable to

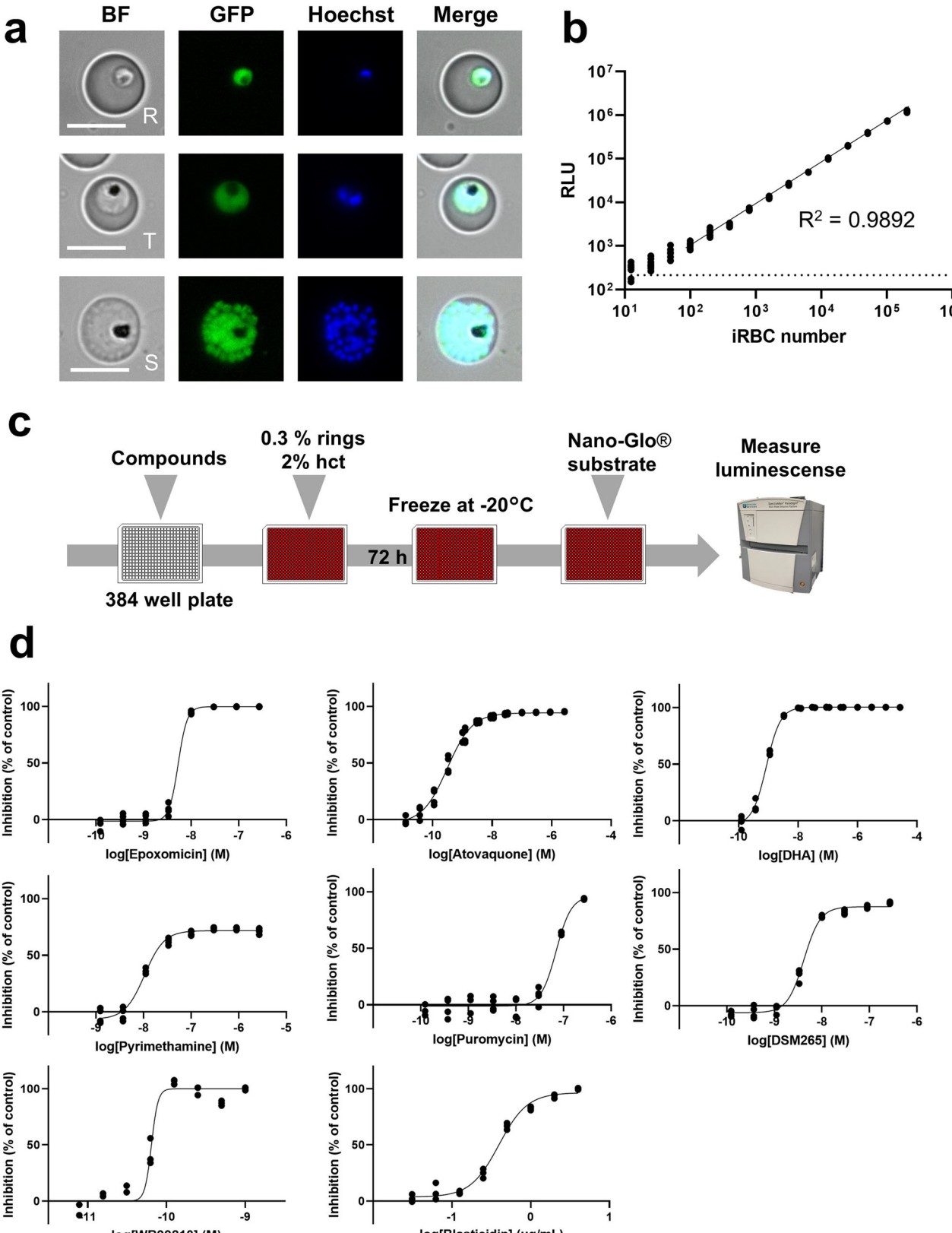

those among NF54 WT parasites (Table 1). On day 7 after feeding, we observed a clear GFP signal in oocysts within the mosquito midgut (Fig. 4a, b). In addition, we detected a strong GFP signal in salivary gland sporozoites (Fig. 4c). To determine whether the parasites could be quantified based on NanoLuc luminescence, we isolated *P. falciparum*-infected midguts and salivary gland sporozoites, and performed a luciferase assay with the lysates of these mosquito samples. A clear correlation between the luminescence intensity and the number of GFP-positive oocysts (Fig. 4d) and sporozoites (Fig. 4e) was observed, thereby indicating that the reporter activity reflects parasite density.

**Fig. 2 Expression of GFP and NanoLuc reporters and establishment of drug assay for the asexual blood stage. a** Representative fluorescence microscopy images of live GFP-NanoLuc parasites in asexual blood stages. R, rings; T, trophozoites; S, schizonts; Nuclei were stained with Hoechst-33342. All images were obtained under standardised exposure/gain times to visualise differences in fluorescence intensity [GFP 0.7 s; Hoechst 0.2 s; bright field 0.1 s (1× gain)]. Live imaging analysis was performed at least thrice. Bright field (BF) Scale bar, 7 μm. **b** Correlation between NanoLuc bioluminescence and the number of the GFP-NanoLuc reporter line parasites in the asexual blood stage. Whole-lysate samples of the GFP-NanoLuc reporter line were prepared via serial dilution. For the NanoLuc reaction, the samples were mixed with Nano-Glo diluted 1:500 and measured in the luminometer. Luciferase activity from NanoLuc is represented as relative light units (RLU). The solid line and the dashed line indicate the linear regression and the background value derived from uninfected RBCs, respectively. Symbols indicate the RLU value from technical eight replicates per dilution. **c** Schematics of the asexual blood stage drug assay. Chemical compounds were dispensed in 384-well white plates and then ring stage-synchronised parasite cultures were applied to each well. After 72 h of incubation at 37 °C, the plates were frozen and reacted with Nano-Glo substrates. **d** Dose-response curves of established antimalarial compounds. The GFP-NanoLuc reporter lines were cultured in different concentrations of indicated anti-malarial compounds, DMSO (negative control well) or 1 μM DHA (positive control well). All data represent the average from technical quadruplicates. Symbols indicate the RLU value per dilution. The IC$_{50}$ value of each anti-malarial compounds was determined via non-linear regression. The IC$_{50}$ values of the anti-malarial compounds are summarised in Table 2.

**Table 2 IC$_{50}$ values in the asexual blood, gametocyte, and liver stage, determined by the GFP-NanoLuc reporter line.**

| | Compound | Incubation time | IC$_{50}$ (±SD) |
|---|---|---|---|
| Asexual stage | Atovaquone | 72 h (384 wells) | 0.31 ± 0.024 nM |
| | Dihydroartemisinin | | 0.86 ± 0.022 nM |
| | Pyrimethamine | | 11 ± 0.66 nM |
| | Puromycin | | 70 ± 4.8 nM |
| | DSM265 | | 4.2 ± 0.16 nM |
| | Epoxomicin | | 5.3 ± 0.27 nM |
| | WR99210 | | 0.064 ± 0.013 nM |
| | Blasticidin | | 0.39 ± 0.021 μg/mL |
| Gametocytes stage III–V | Epoxomicin | 72 h + 0 h[a] | 4.6 ± 2.0 nM |
| | | 72 h + 24 h[b] | 3.7 ± 1.4 nM |
| | | 72 h + 48 h[c] | 3.3 ± 1.4 nM |
| | | 72 h + 72 h[d] | 3.8 ± 1.8 nM |
| | | 72 h (384 wells)[e] | 8.1 ± 0.51 nM |
| | Atovaquone | 72 h + 0 h[a] | > 1000 nM |
| | | 72 h + 24 h[b] | > 1000 nM |
| | | 72 h + 48 h[c] | > 1000 nM |
| | | 72 h + 72 h[d] | > 1000 nM |
| | | 72 h (384 wells)[e] | > 1000 nM |
| | Dihydroartemisinin | 72 h (384 wells)[e] | > 1000 nM |
| Liver stage (NanoLuc) | Atovaquone | 4 dpi | 12 ± 5.1 nM |
| | | 5 dpi | 11 ± 2.7 nM |
| | | 6 dpi | 5.9 ± 1.9 nM |
| Liver stage (No. of schizonts) | | 4 dpi | 17 ± 7.2 nM |
| | | 5 dpi | 17 ± 5.8 nM |
| | | 6 dpi | 7.4 ± 1.7 nM |
| Liver stage (total area of parasites) | | 4 dpi | 13.9 ± 6.2 nM |
| | | 5 dpi | 13.0 ± 7.8 nM |
| | | 6 dpi | 6.1 ± 1.4 nM |

The IC$_{50}$ values of each test compound were calculated from technical triplicate or quadruplicate using GraphPad Prism 9.0 software.
[a–d]After 72 h incubation with the compound, the medium was replaced with fresh one, and NanoLuc assay with extracellular NanoLuc inhibitor was immediately performed[a], or after 24 h[b], 48 h[c], or 72 h[d] additional incubation.
[e]After 72 h incubation with the compound, NanoLuc assay with extracellular NanoLuc inhibitor was immediately performed.

**Analysis of reporter expression and validation of drug assay using the GFP-NanoLuc reporter line at the liver stage.** To characterise reporter expression at the liver stage, we isolated salivary gland sporozoites of the GFP-NanoLuc line and performed an infection experiment using primary human hepatocytes. First, live imaging analysis revealed GFP-positive liver-stage parasites (Fig. 5a), which was consistent with previously observed reporter expression in primary hepatocytes[16]. Compared with NF54 WT parasites, the GFP-NanoLuc line showed equal or increased infectivity to the same batch of primary human hepatocytes and a comparable increase in size between days 3 and 9 post-inoculation of sporozoites into the cells (Fig. 5b, c). Phosphatidylinositol 4-kinase inhibitor MMV390048 and atovaquone blocked the luminescence signal at day 4, 5 and 6 post infection, in keeping with their inhibitory effect on development of liver schizonts[42] (Fig. 5d). Assay parameters (%CV$_{max}$; %CV$_{min}$; S/N; S/B; Z') were in general better when calculated for the difference between control (0.1% DMSO) and MMV390048 treated wells than for the atovaquone conditions, but below 0.5 under all conditions (Table 3). Atovaquone dose-dependently reduced luminescence signals in infected hepatocytes with IC$_{50}$ values in line with previously reported values (Table 2)[12,14,43]. The luminescence-based assay does not discriminate between the number of infected hepatocytes and the size of the developing parasites. Nevertheless, we observed near identical IC$_{50}$ values when parasitemia was determined by imaging and quantification of the number of HSP70 positive forms, the total area of HSP70 staining or the Nano-Glo-derived luminescence signals (Fig. 5e–g, Table 2). Overall, we observed a high degree of correlation between luminescence values and number of liver stage schizonts (Fig. 5h; R$^2$ = 0.90; 0.95; 0.96 for day 4, 5 and 6, respectively). In line with the larger size of the schizonts, luminescence values were higher at day 5 and 6 post infection in comparison with day 4 parasites. The combined data indicate that the GFP-NanoLuc reporter line can be a novel and powerful tool to evaluate the potency and efficacy of anti-malarial compounds against liver stage parasites.

**HTS of anti-malarial compounds using the Osaka University chemical library.** To discover novel anti-malarial compounds against multi-stage *P. falciparum*, we performed HTS using the GFP-NanoLuc reporter line (Fig. 6a). First, 1920 compounds from the Osaka University chemical library were screened at 2 μM using the asexual blood stage GFP-NanoLuc reporter line, of which, 30 compounds showing >80% inhibition of asexual parasite growth were subjected to subsequent gametocytocidal assays (Fig. 6b). The Z'factor of every plate was >0.8 (Fig. 6c), confirming the robustness of the drug assays during HTS. For the gametocytocidal assays, we applied the 72 h + 0 h protocol with the extracellular NanoLuc inhibitor in a 96-well plate format considering that it exerted high assay quality and required a short time (Table 3). OU0074008 achieved 20% gametocyte inhibition at 4 μM (Fig. 6d). We determined the IC$_{50}$ values of OU0074008 against the asexual blood stage (Fig. 6e) and gametocyte stage (Fig. 6f), which confirmed its anti-malarial activity against asexual

**Table 3 Parameters of the drug assays using the GFP-NanoLuc reporter line at asexual blood and gametocyte stages.**

| | Asexual stage | Gametocyte stage | | | | | | | | Liver stage | | | | | |
| | 384 wells | 96 wells | | | | | | | 384 wells | 96 wells | | | | | |
| | | NanoLuc inhibitor (−) | NanoLuc inhibitor (+) | | | | | | | DMSO-Atovaquone[g] | | | DMSO-MMV390048[h] | | |
| | 72h | 72h[a] | 72h[b] | 72h + 0h[c] | 72h + 24h[d] | 72h + 48h[e] | 72h + 72h[f] | | 72h[b] | 4 dpi | 5 dpi | 6 dpi | 4 dpi | 5 dpi | 6 dpi |
| | (n = 13) | exp.1 | exp.1 | exp.1 (n = 4) | exp.1 | exp.1 | exp.1 | exp.2 | exp.1 | exp.1 | exp.1 | exp.1 | exp.1 | exp.1 | exp.1 |
| Z'factor | 0.88 ± 0.02 | 0.43 | 0.89 | 0.80 ± 0.08 | 0.95 | 0.81 | 0.91 | 0.75 | 0.65 | −0.12 | 0.04 | −0.73 | 0.02 | 0.06 | −0.71 |
| S/B | 41.3 ± 5.9 | 1.8 | 3.4 | 6.9 ± 1.3 | 9.1 | 17.0 | 21.2 | 14.5 | 8.4 | 8.1 | 33.2 | 46.6 | 12.1 | 52.6 | 66.7 |
| S/N | 484 ± 71 | 14 | 54 | 129 ± 14 | 107 | 223 | 248 | 107 | 126 | 28 | 272 | 292 | 220 | 376 | 1130 |
| %CV$_{max}$ | 3.6 ± 0.6 | 5.2 | 1.4 | 4.9 ± 2.2 | 0.5 | 5.5 | 2.4 | 7.2 | 9.7 | 29.6 | 30.6 | 56.0 | 5.1 | 13.7 | 5.8 |
| %CV$_{min}$ | 8.4 ± 1.4 | 5.8 | 4.4 | 4.6 ± 1.1 | 7.6 | 7.2 | 8.1 | 8.2 | 5.9 | 25.5 | 11.8 | 15.6 | | | |

a After 72h incubation with compounds, the same NanoLuc assay protocol as that for asexual blood stage parasites was performed.
b After 72h incubation with compounds, NanoLuc assay with extracellular NanoLuc inhibitor was immediately performed.
c,f After 72h incubation with compounds, the medium was replaced with fresh one and NanoLuc assay with extracellular NanoLuc inhibitor was immediately performed[c], or after 24 h[d], 48 h[e], or 72 h[f] additional incubation.
g All values were calculated when atovaquone was defined as a positive control.
h All values were calculated when MMV390048 was defined as a positive control.
exp, the number of biological replicates.

blood stages. OU0074008 exhibited *P. falciparum*-asexual blood stage specific cytotoxicity, as its IC$_{50}$ value against human cells (HepG2 cells) was higher than that against the asexual blood-stage parasite (Fig. 6g). To verify the anti-malarial activity of OU0074008, we performed a lactate dehydrogenase (LDH) assay and confirmed that the IC$_{50}$ value from different types of viability assay was equivalent to that from the NanoLuc assay (Supplementary Fig. 5). Furthermore, the NanoLuc inhibition assay with OU0074008 suggests that there is no obvious NanoLuc inhibitory activity of the hit compound (Supplementary Fig. 5). Though OU0074008 is effective against asexual blood-stage parasite, the compound did not show inhibitory activity against liver stage *P. falciparum* parasites (Supplementary Fig. 6). Altogether, our HTS using the GFP-NanoLuc reporter line provides an experimental framework for identifying anti-malarial compounds and demonstrates the utility of this reporter line for multi-stage drug assays.

## Discussion

Herein, we generated a versatile *P. falciparum* transgenic parasite expressing NanoLuc, a superbright luciferase which allows for parasite quantification. Since the GFP-NanoLuc reporter line expresses two different reporter proteins throughout the whole life cycle, this transgenic line would be a valuable molecular tool to visualise parasites and evaluate anti-malarial drug efficacy at the asexual blood, gametocyte, oocyst, sporozoite, and liver stages. As a proof of concept, we used the GFP-NanoLuc reporter line for HTS against asexual blood stage parasites and gametocytes, identifying OU0074008 as a novel anti-malarial compound.

We propose that the GFP-NanoLuc reporter line generated in this study would have an advantage for the quantification of multi-stage parasites when compared to several previously established reporter lines, owing to its greater brightness when compared to firefly luciferase[33,44]. Although several firefly luciferase-expressing *P. falciparum* reporter lines have been generated for multi-stage analysis[14,17,18,27], established NanoLuc transgenic reporter lines have not been applied for evaluating anti-malarial compounds against gametocytes, mosquito-stage, and liver-stage parasites[28,30,45,46]. The first *P. falciparum* 3D7 transgenic line expressing an exported NanoLuc was extensively used for testing anti-malarial activity against asexual blood-stage parasites and was applied for screening inhibitors of protein transport and invasion/egress[30,45,46]. In this transgenic line, NanoLuc expression is controlled by a constitutive promoter in an episomal plasmid with a drug selection marker, which could lead to the loss of the plasmid in mosquito stages, wherein drug pressure cannot be applied. Hence, the ideal approach for NanoLuc expression at multiple stages is the marker-free integration of the NanoLuc cassette into the genome of gametocyte-producing strains, such as NF54. A transgenic line, wherein NanoLuc expression is driven by the ef-1α promoter in the NF54 strain, was recently established[28]. However, the application of this transgenic reporter line has thus far been limited to the asexual blood stage[28]. In our study, we showed that luminescence and parasite number are well correlated, proposing that luminescence from a constitutive promoter would be a reasonable readout for quantifying *P. falciparum* parasites across asexual blood, gametocyte, oocyst, sporozoites, and liver stages. Moreover, immunocompromised mice engrafted with human red blood cells and primary human hepatocytes were established and is now an instrumental resource for preclinical drug and vaccine safety and efficacy screens. For in vivo studies using humanized mice, the GFP-NanoLuc reporter line generated in this study may be a valuable tool to investigate how the parasite grows and evaluate anti-malarial compounds in the infection model.

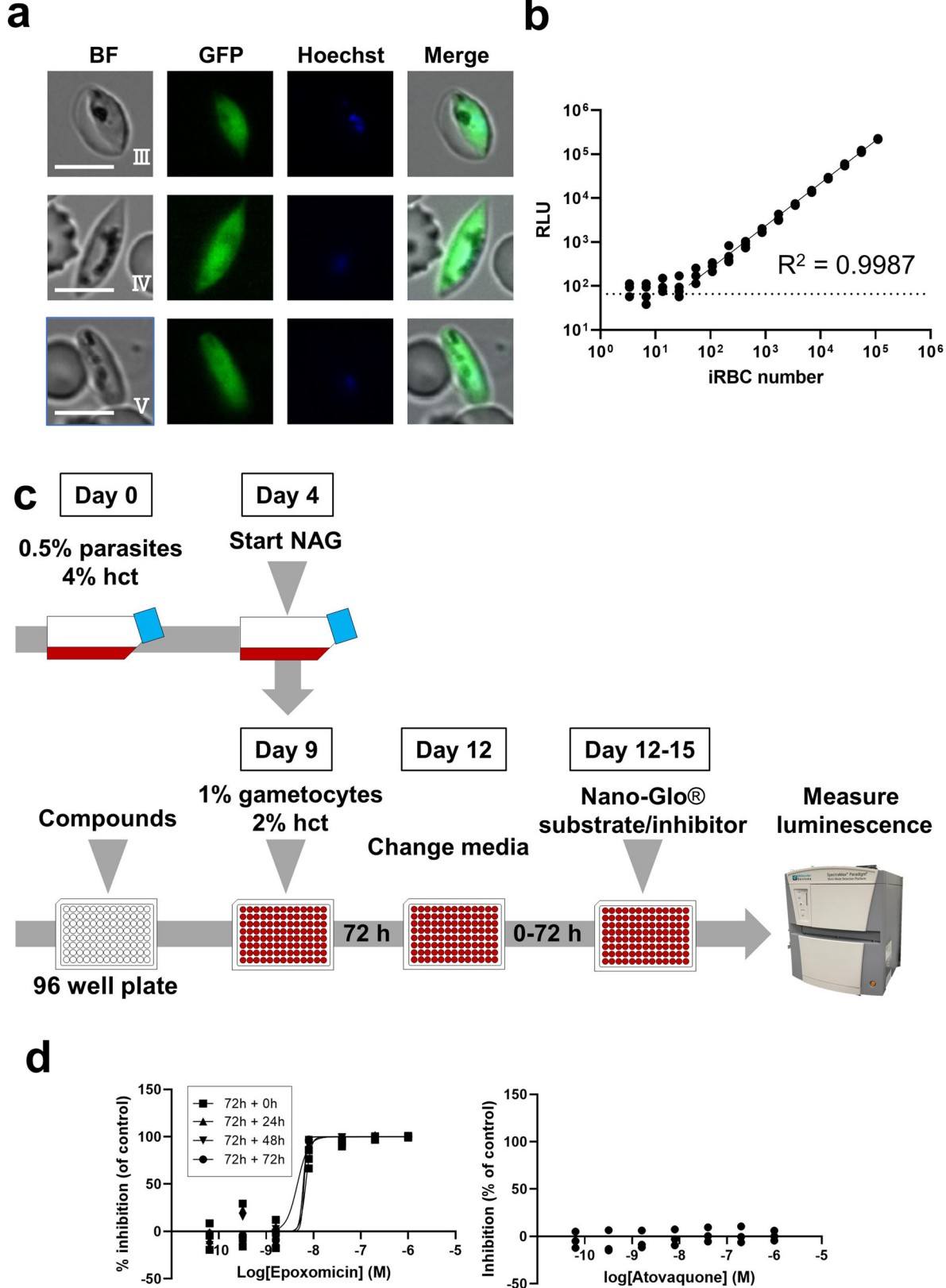

The *P. falciparum* GFP-NanoLuc reporter line can be used for HTS against the asexual blood and gametocyte stages with highly robust drug assay parameters. For drug assays against asexual blood- and gametocyte-stage *P. falciparum*, we optimised assay conditions in 96- or 384-well plates and examined the efficacy of known anti-malarial compounds, highlighting the utility of our reporter line for HTS. Moreover, our *P. falciparum* transgenic reporter line is marker-free and was generated via CRISPR/Cas9-induced double crossover recombination. As expected, the transgenic reporter line was sensitive to two anti-malarial drugs, WR99210 and blasticidin, used for positive drug selection. The absence of a drug selection marker would avoid

**Fig. 3 Expression of GFP and NanoLuc reporters and establishment of drug assay for the gametocyte stage. a** Representative fluorescence microscopy images of live GFP-NanoLuc line parasites in gametocyte stages III, IV, and V. Nuclei were stained with Hoechst-33342. All images were captured with standardised exposure/gain times to visualise differences in fluorescence intensity [GFP 0.7 s; Hoechst 0.2 s; bright field 0.1 s (1× gain)]. Live imaging analysis was performed at least thrice. Bright field (BF) Scale bar, 7 μm. **b** Correlation between NanoLuc bioluminescence and the number of the GFP-NanoLuc reporter cells at the gametocyte stage. The whole-lysate samples of GFP-NanoLuc gametocytes were prepared by serial dilution. For the NanoLuc reaction, samples were mixed with Nano-Glo diluted 1:500 and measured in the luminometer. Luciferase activity from NanoLuc is represented as RLU. The solid and dashed lines indicate the linear regression and the background value derived from uninfected RBCs, respectively. Symbols indicate the RLU value from quadruplicates per dilution. **c** Schematics of gametocyte stage drug assay using the GFP-NanoLuc reporter line. Chemical compounds were dispensed in 96-well white plates, and gametocyte cultures were then applied to each well. After 72 h of incubation at 37 °C, the old medium was replaced with fresh medium, and then plates used for reaction with Nano-Glo substrates after additional incubation ranging from 0–72 h. The extracellular NanoLuc inhibitor was used to avoid bioluminescence from residual NanoLuc released from dead gametocytes. **d** Dose-response curves of epoxomicin and atovaquone for gametocytes. The GFP-NanoLuc reporter lines were cultured in different concentrations of the indicated anti-malarial compounds, DMSO (negative control well) or 1 μM epoxomicin (positive control well). Symbols indicate the RLU value from technical triplicates per dilution. The IC$_{50}$ of epoxomicin was determined via non-linear regression. The IC$_{50}$ values of anti-malarial compounds are summarised in Table 2.

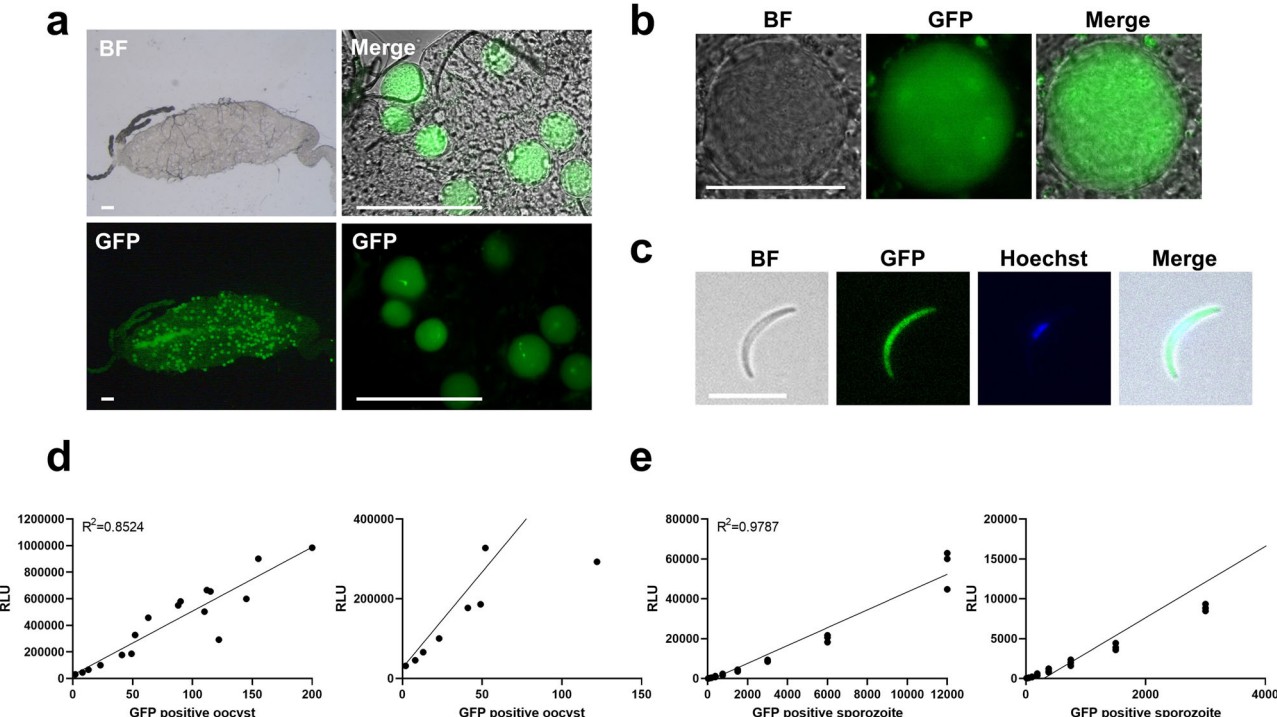

**Fig. 4 Expression of GFP and NanoLuc reporters at the mosquito stage. a** Representative fluorescence microscopy pictures of *Anopheles* mosquito midgut infected with the GFP-NanoLuc reporter line. The oocysts in the complete midgut are shown as puncta in the image. The complete midgut is shown on the left, and zoomed images of the oocysts are shown on the right. Live imaging analysis was performed at least thrice. Scale bar, 40 μm. **b** Representative images of live oocysts of the GFP-NanoLuc line in *A. stephensi* mosquitoes. Scale bar, 10 μm. **c** Representative fluorescence microscopy images of live salivary gland sporozoites of the GFP-NanoLuc reporter line isolated on day 24 after mosquito infection. Nuclei were stained with Hoechst-33342. Bright field (BF). Scale bar, 7 μm. **d**, **e** Correlation between NanoLuc bioluminescence and the number of the GFP-NanoLuc reporter cells in the oocyst (**d**) or sporozoite stages (**e**). Left panel; Overall correlation, Right panel; Zoomed correlation. Whole-midgut samples or whole-lysate sporozoite samples were used for the NanoLuc reaction. For the reaction, samples were mixed with Nano-Glo diluted at 1:500 and measured in the luminometer. Luciferase activity from NanoLuc is presented as RLU. Each dot indicates the RLU from a single midgut sample in (**d**). Symbols indicate the RLU value from technical triplicates per dilution in (**e**). The solid lines indicate the linear regression.

potential confounding resistance against anti-malarial compounds tested in the HTS. Notably, the gametocytocidal assay with the GFP-NanoLuc line is a simple and user-friendly system that enables HTS with high sensitivity in any laboratory settings as long as standard equipment is prepared. Gametocyte purification with a magnet or percoll to remove the effect of background derived from uninfected RBCs and IFA to visualize parasites, both of which lower the throughput significantly, is not necessary for the gametocytocidal assay with the GFP-NanoLuc line. In addition, expensive imaging devices to quantify parasite numbers, such as a high-content imaging system,

are not needed. Furthermore, it is theoretically possible to investigate the antimalarial effect of compounds at the early gametocyte stage, which is achieved by shortening the culture time post-gametocyte induction, although we did not perform an assay with early-stage gametocytes in this study. In this case, the purification of stage I gametocytes may be required before seeding the parasites on an assay plate. The only limitation of the gametocytocidal assay with the GFP-NanoLuc line is that the costs for the extracellular NanoLuc inhibitors provided by Promega are relatively high, which is not suitable for high-throughput screening with limited resources.

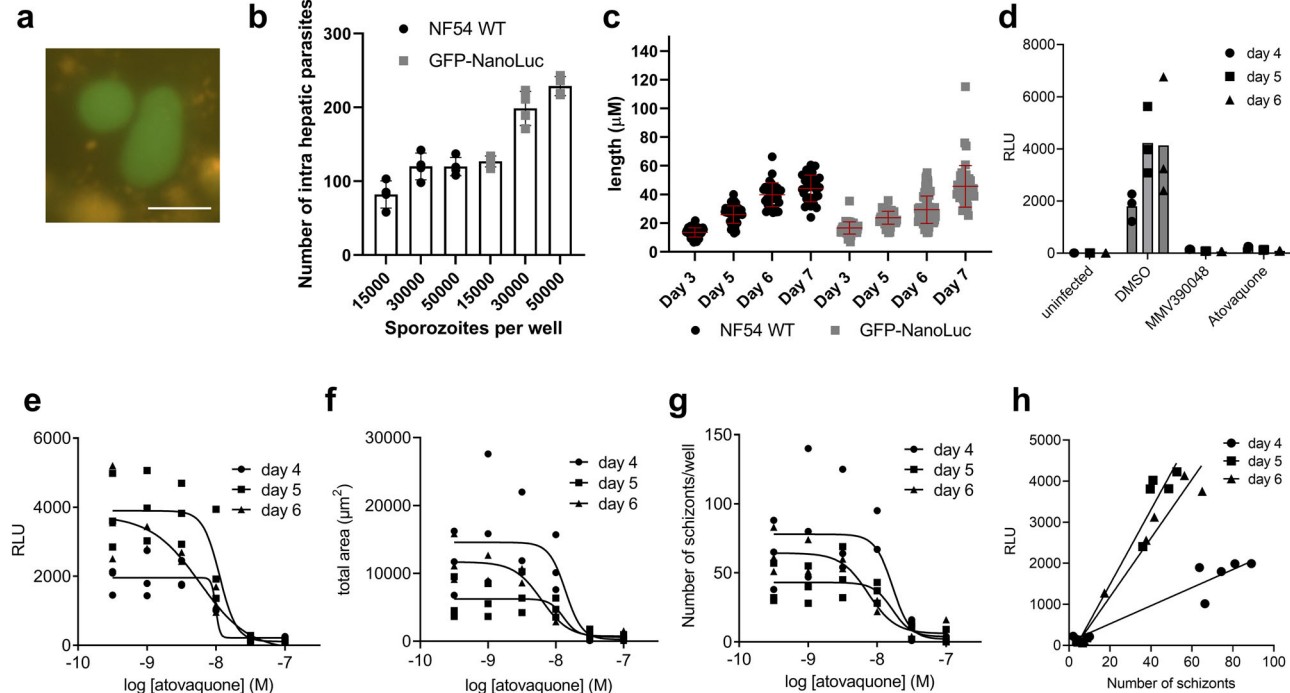

**Fig. 5 Evaluation of antimalarial drug efficacy using the GFP-NanoLuc reporter line at the liver stage. a** Representative live fluorescence microscopy picture of GFP-NanoLuc liver-stage parasites at day 7 in primary human hepatocytes. Live imaging analysis was performed at least thrice. Scale bar, 50 μm. **b** Numbers of NF54 WT and GFP-NanoLuc intrahepatic parasites on day 7 of development within primary human hepatocytes according to the number of sporozoites inoculated to the cells. Means and SD from technical quadruplicates are shown as box bars and error bars, respectively. **c** Length of NF54 WT and GFP-NanoLuc exoerythrocytic form (EEF) parasites at different time points of development within primary human hepatocytes. Each dot indicates the individual length of a single EEF parasite. Means and SD of the parasite length on each time point (containing each 50–100 intrahepatic parasites) are shown as a horizontal line and an error bar, respectively. **d** Inhibition of intrahepatic parasite invasion and/or development by treatment with a single dose of 100 nM Atovaquone or 1 μM MMV390048 on day 4, 5, or 6 post infection. Means and SD from technical triplicates are shown as box bars and error bars, respectively. **e–g** Atovaquone dose-dependent inhibition of intrahepatic parasite invasion and/or development assessed either by luminescence (**e**) or imaging of HSP70 positive forms by total surface area (**f**) or number of infected hepatocytes (**g**) on day 4, 5 or 6 post infection. The total area calculated by the image software in (**f**) is defined as DAPI positive total surface area of both hepatocyte and schizont nuclei for HSP70 positive forms where the HSP70 stain overlaps any of the DAPI stains. Luciferase activity from NanoLuc is represented as RLU. Symbols indicate the values from technical triplicates per dilution. (**h**) Correlation between NanoLuc luminescence and the number of EEF in the primary human hepatocytes. Each dot indicates the RLU value from a single well which contains the GFP-NanoLuc line-infected hepatocytes.

The GFP-NanoLuc reporter line would be valuable for the evaluation of anti-malarial compounds against liver-stage *P. falciparum*. Currently, the analysis of *P. falciparum* liver-stage development remains challenging due to several technical limitations, such as the low infectivity of *P. falciparum* sporozoites and parasite tropism for human hepatocytes[2]. Herein, we report that the GFP-NanoLuc liver-stage parasites developing in primary human hepatocytes express GFP and show similar or higher infection rate and growth when compared to that of NF54 parasites. These characteristics could be suitable for visualising the liver-stage parasites and sorting the infected hepatocytes. To evaluate the efficacy of the anti-malarial compounds atovaquone and MMV390048, we performed drug assays using a GFP-NanoLuc reporter line and primary human hepatocytes. Moreover, we demonstrated that the activity of atovaquone and MMV390048 against liver stage *P. falciparum* in primary human hepatocytes can be evaluated using our drug assay system with brighter luminescence when compared to a previously established system using Firefly luciferase[14]. Although throughput of drug assay against liver-stage parasites is relatively low when compared to that against the blood-stage parasites, multiple compounds could be tested in our 96-well-based drug assay format.

We used the GFP-NanoLuc reporter line for the HTS of 1920 compounds comprising drug-like scaffolds and identified a novel compound that appeared to be effective against both asexual blood- and gametocyte-stage parasites. Over 30 compounds were effective against the asexual blood stage, among which, OU0074008 appeared to exhibit modest anti-gametocyte efficacy with an IC50 value of 12 μM. Based on the identification of OU0074008, we propose that our HTS platform can aid in the identification of novel anti-malarial compounds. Although the IC50 value against gametocytes was relatively high, a unique tetracyclic silicon-containing cyclic compound OU0074008 would be one of the potential candidates for further drug discovery, since various derivatives of OU0074008 can be easily and efficiently synthesized from the corresponding precursor, ortho-alkynylphenyl allyl dimethylsilane, *via* a one-pot Enyne Metathesis/Diels–Alder reaction. OU0074008 is also unique because it has two kinds of lipophilic elements, silicon and fluorine. Another interesting factor is the possibility to further improve the desired biological activity, while confirming where these elements are important in terms of activity statements in the basic skeleton[47].

In conclusion, we generated a novel transgenic *P. falciparum* reporter line expressing GFP and NanoLuc throughout its life cycle. Based on luminescence derived from the reporter line, we established a robust drug assay protocol which can be used for asexual blood-, gametocyte-, and liver-stage parasites. Such versatile transgenic reporter parasites will boost the identification of the novel class of anti-malarial compounds effective against parasites in the human host and preventing human–mosquito transmission.

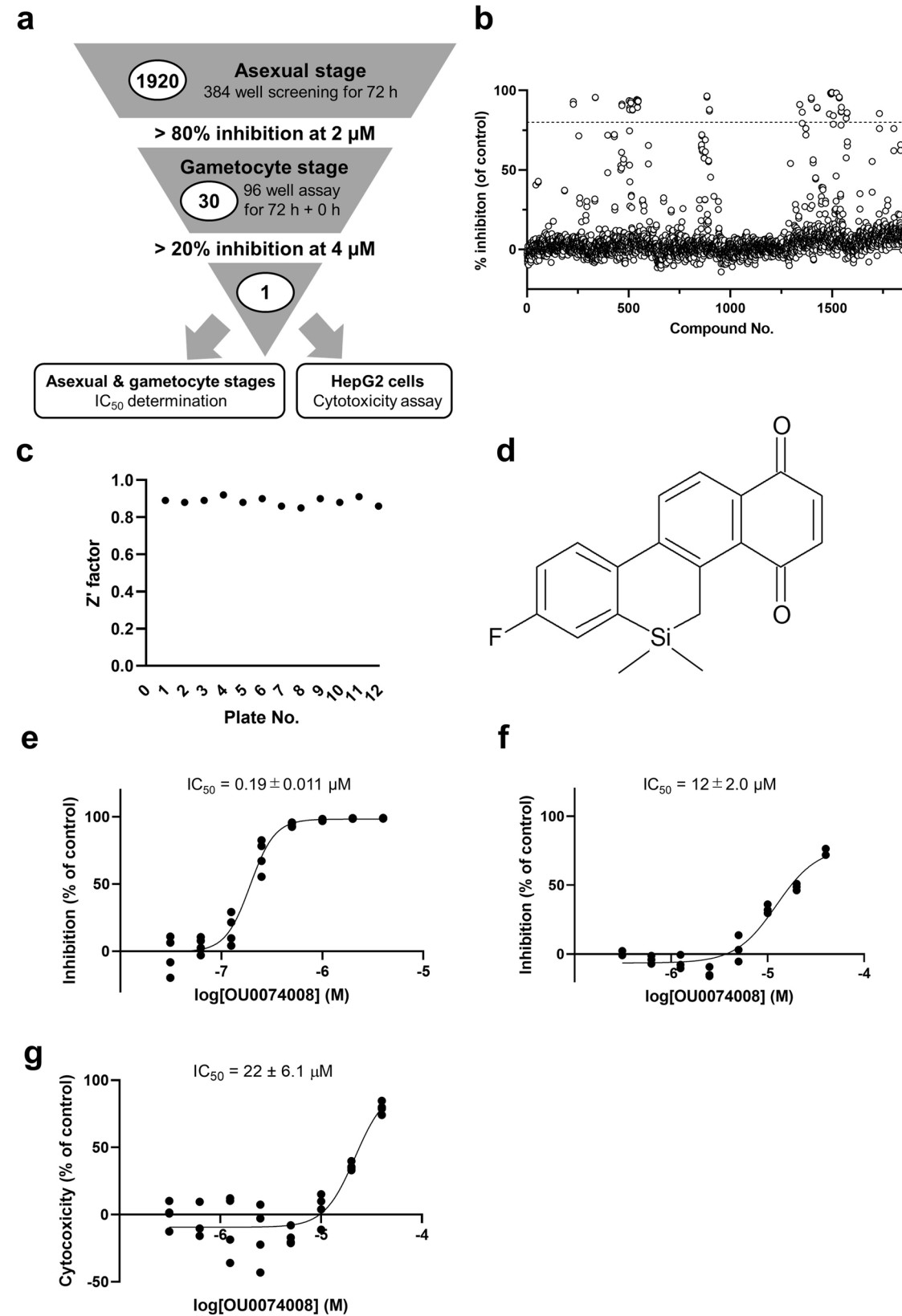

## Methods

**In vitro cultivation of asexual blood-stage *P. falciparum*.** *P. falciparum* NF54 strain and its transgenic lines were maintained in a standard semi-automated shaker condition at Leiden University[15]. Fresh human serum and human red blood cells (RBCs) were obtained from the Dutch National Blood Bank (Sanquin Amsterdam, the Netherlands; permission granted from donors for the use of blood products for malaria research and microbiology tested for safety). RBCs of different donors were pooled every two weeks, washed twice in serum-free RPMI-1640, and suspended in a complete culture medium to 50% haematocrit. Human serum samples from different donors were pooled every 4–6 months and stored at −20 °C until the required. *P. falciparum* strains were cultured under static conditions at Nagasaki University, as previously described[48]. Briefly, parasites were cultured in RPMI-1640 medium (Gibco) containing O+ RBCs at 4% haematocrit supplemented with 0.5% (w/v) AlbuMAX I (Invitrogen), 200 μM hypoxanthine (Sigma),

**Fig. 6 Screening of 1,920 compounds from the Osaka University chemical library using the GFP-NanoLuc reporter line. a** Schematics of Osaka University chemical library drug screening. Screening against asexual blood-stage parasites was performed at a final concentration of 2 μM. The 30 hit compounds were subsequently screened at the gametocyte stage at a final concentration of 4 μM. One hit compound was further validated via determination of IC$_{50}$ value against asexual blood stage, gametocyte stage and human HepG2 cells. **b** Overview of the screening of 1,920 compounds against the asexual blood stage. GFP-NanoLuc line was cultured in 2 μM of each compound, DMSO (negative control well) or 1 μM DHA (positive control well) on 384-well plates. The dashed line represents the 80% inhibition threshold for the selection of hit compounds. The assay was performed in technical duplicate. **c** Z'factor of the screening for asexual blood stage parasites (total 12 assay plates). **d** Chemical structure of OU0074008, a hit compound effective against asexual blood and gametocyte stage. **e** Determination of IC$_{50}$ value of the hit compound OU0074008 against the asexual blood stage. The GFP-NanoLuc reporter line was cultured in different concentrations of OU0074008, DMSO (negative control well) or 1 μM DHA (positive control well) on 96-well plates. The IC$_{50}$ value of OU0074008 was determined via non-linear regression from technical quadruplicate using GraphPad Prism 9.0 software. **f** Determination of IC$_{50}$ value of the hit compound OU0074008 against the gametocyte stage. The GFP-NanoLuc reporter line was cultured in different concentrations of OU0074008, DMSO (negative control well) or 1 μM epoxomicin (positive control well) on 96-well plates. The IC$_{50}$ value of OU0074008 was determined via non-linear regression from technical triplicate using GraphPad Prism 9.0 software. **g** Cytotoxicity of OU0074008 against HepG2 cells. The cells were exposed for 72 h with indicated concentration of OU0074008 or DMSO (control well) on a 96 well plate and cytotoxicity assay using CCK-8 was performed. The IC$_{50}$ value of OU0074008 was determined via non-linear regression from technical quadruplicate using GraphPad Prism 9.0 software.

and 10 μg/mL gentamicin (Sigma). Human RBCs and plasma were obtained from the Nagasaki Red Cross Blood Center (Nagasaki, Japan; permission granted from donors for the use of blood products for malaria research and microbiology tested for safety). RBCs and human serum samples from different donors were pooled. Cloning of the GFP-NanoLuc reporter line was performed via limiting dilution, as previously described[15]. The cultures were diluted to a final parasitaemia of 0.1% and maintained with daily medium changes to monitor the growth of GFP-NanoLuc reporter parasites. Parasitaemia in NF54 WT and GFP-NanoLuc cells was determined via Giemsa staining.

**P. falciparum gametocyte production**. For mosquito feeding, gametocytes were produced using the standard crash method, as previously described[49,50]. Briefly, the asexual blood-stage parasites were diluted to a final parasitaemia of 0.5%, and cultures were maintained with a daily medium change for 14–17 days without replenishing fresh RBCs.

For the NanoLuc assay, gametocyte induction was initiated as described below, with slight modifications. Briefly, the GFP-NanoLuc reporter line was diluted to 0.5% parasitaemia at 4% haematocrit on day 0, and the medium was changed daily until each analysis. Fifty mM N-acetylglucosamine (NAG) (Sigma) was added to the gametocyte culture from day 4 to eliminate asexual blood-stage parasites. Gametocyte production was confirmed via Giemsa staining.

**Plasmid construction**. Donor plasmids used in this study are shown in Supplementary Fig. 2a. The DNA sequence of *P. falciparum* codon-optimised NanoLuc fused with T2A and part of the GFP ORF (the complete sequence is shown in Supplementary Fig. 2b) was synthesised by Integrated DNA Technologies (IDT, gBlocks®). To construct the donor plasmid harbouring the GFP-T2A-NanoLuc expression cassette, we first PCR-amplified (KOD Hot Start DNA Polymerase, Merck Millipore) the sequence containing partial GFP, T2A, and full NanoLuc using primers **P9/P10** (Supplementary Table 1). The amplified GFP-T2A-NanoLuc sequence was introduced into the BtgZI/AvrII site of pLf0127 (*p47* gRNA/*p47* HR1-HR2 GFP) using the InFusion reaction (In-Fusion® HD Cloning Kit; Clonetech), which was previously used to generate a transgenic line expressing GFP[16]. To construct the mCherry-T2A-NanoLuc expression cassette, we PCR-amplified the T2A-NanoLuc using primers **P11/P10** (Supplementary Table 1). The amplified PCR fragment was introduced into the EagI/AvrII site of pLf0128 (*p47* gRNA/*p47* HR1-HR2 mCherry-Luc) by an InFusion reaction (In-Fusion® HD Cloning Kit, Clontech) to replace firefly luciferase with the T2A-NanoLuc coding sequence. The donor/gRNA plasmid for mCherry-Luc (pLf0128) has been described previously[16].

**Generation of *P. falciparum* transgenic parasites**. To create the reporter lines, we used a previously described Cas9 construct (pLf0019), containing the Cas9 expression cassette with a blasticidin (BSD) drug-selectable marker cassette[15] in combination with a gRNA/Donor DNA plasmid (pLf0181 and pLf0182). The gRNA-donor DNA constructs contain hdhfr-yfcu drug-selectable marker (SM) cassette for selection with the drug WR99210. In addition, it contains two homologous regions targeting *p47* (PF3D7_1346800) and gRNA targeting the *p47* locus (gRNA019) from the plasmid pLf0047[14]. Transfection was performed using the spontaneous uptake method, as previously described[51,52]. Briefly, CRISPR constructs (a mixture of ~50 μg of each circular plasmid) were first introduced into freshly isolated RBCs using a Gene Pulser Xcell electroporator (BioRad) with a single pulse (310 V, 950 μF, and ∞ capacity). To generate the two different reporter lines, the Cas9 expression plasmid (pLf0019) and donor/gRNA plasmids (pLf0181 for mCherry-NanoLuc or pLf0182 for GFP-NanoLuc) were simultaneously transfected into parasites. Subsequently, plasmid-loaded RBCs were incubated with mixed stages of *P. falciparum*-infected RBCs with 0.5% parasitemia and cultured in vitro for three days. Once parasitaemia reached 5%, double drug selection with

WR99210 (2.6 nM, Jacobus Pharmaceutical) and BSD (5 μg/mL, Sigma) was applied for 6 days to obtain transgenic parasites with the integration of the donor sequence. Positive selection helped in identifying the parasites that were transfected successfully with both plasmids (Cas9 and gRNA/Donor DNA constructs). Cultures were maintained in drug-free media until thin blood-smears were parasite-positive (usually after 14–21 days). After double drug selection, the transfectants were selected using 5-fluorocytosine (1 μM, Ancotil, MEDA pharma) to obtain the double cross-over parasites. Subsequently, GFP-positive transgenic parasites were enriched via flow cytometry-based sorting, as previously described[16]. Clonal lines of GFP-NanoLuc were isolated through limiting dilution. Cloned parasites were transferred in 10 mL culture flasks at 5% haematocrit and cultured under standard culture conditions in a semi-automated culture system for collection of parasites for further genotype and phenotype analyses (see next section). Two independent GFP-NanoLuc cell lines were obtained (Exp245 and Exp227). A clonal GFP-NanoLuc line (Exp245) was used in most experiments. Exp227 was used for the NanoLuc reaction at the oocyst and sporozoite stages (Supplementary Fig. 3).

**Genotyping of the transgenic parasites**. Genomic DNA was extracted for genotyping using a Wizard® Genomic DNA Purification Kit (Promega) or phenol/chloroform, as previously described[15]. The DNA fragments were amplified by PCR using CloneAmp HiFi PCR premix (Clontech) or KOD hot-start DNA polymerase (Merck Millipore), according to the manufacturer's instructions. For amplification by KOD hot-start DNA polymerase, the DNA fragments were PCR-amplified under standard conditions at annealing temperatures of 50, 55, and 60 °C for 10 s and an elongation step of 68 °C. The amplified PCR products were visualised using GelRed Nucleic Acid Gel Stain (Biotiun) or ethidium bromide (Sigma).

**Western blotting**. For Western blot analysis of GFP-NanoLuc expression, total protein extracts from in vitro-cultured mixed asexual blood stage *P. falciparum* parasites (line Exp245 clone1) were separated via 12% (w/v) SDS-PAGE gel and transferred to a PVDF transfer membrane (Amersham Hybondtm-P) by electroblotting. GFP-NanoLuc expression was detected by incubating the membrane with rabbit polyclonal anti-GFP antibody (1:1000; Abcam; ab290), followed by incubation with a horseradish peroxidase (HRP)-conjugated goat anti-rabbit IgG secondary antibody (1:10,000; GE Healthcare; NA934V). Immunostained proteins were visualised by incubating the membrane with Pierce ECL Plus Substrate (Thermo Scientific; 32132). The resulting chemiluminescent signals were captured using X-ray film (SuperRX-N Fuji Medical).

**Live imaging using fluorescence microscopy**. Fluorescent protein expression in the asexual blood and gametocyte stages was analysed using standard fluorescence microscopy, as previously described[16]. Briefly, 200–300 μL samples of *P. falciparum*-infected RBCs were collected from the 10 mL cultures with parasitaemia between 4–10% and stained with a DNA-specific dye, Hoechst-33342 (Thermo Fisher) by adding 4 μL of a 500 μM stock solution at a final concentration of 10 μM for 20 min at 37 °C. Subsequently, 10–20 μL of the solution was placed on a microscope slide mounted under a coverslip and fluorescence of live infected RBCs was analysed using a Leica fluorescence MDR microscope (×100 magnification). Images were captured with a DC500 digital camera using ColourPro software with the following exposure times: GFP 0.7 s; mCherry 1 s; Hoechst 0.2 s; bright field 0.1 s (1× gain).

For analysing reporter expression in mosquito-stage parasites (oocysts and sporozoites), *A. stephensi* mosquitoes were infected with day 14–17 gametocyte cultures using the standard membrane feeding assay (SMFA)[53,54]. The transgenic reporter lines were dissected and subjected to imaging via fluorescence microscopy, as previously described[16]. For the oocyst stage, midgut samples isolated on days 8 or 10 after feeding were placed on a microscope slide mounted with a coverslip.

GFP expression was analysed using a Leica fluorescence MDR microscope (100× magnification). Images were captured with a DC500 digital camera microscope using Leica LAS X software at the following exposure times: GFP 0.7 s; Hoechst-33342 0.136 s; bright field 0.62 s (1× gain). Collection of salivary gland sporozoites for counting numbers and GFP expression was performed 24 days after feeding. The salivary gland sporozoites were isolated for GFP expression by centrifugation ($800 \times g$, 5 min). The pellet was suspended in 40 μL PBS, and sporozoites were stained with Hoechst-33342 (10 μM) for 30 min at 37 °C. Of this solution, 5–10 μL was placed on a microscopic slide mounted under a coverslip, and the GFP fluorescence of live sporozoites was analysed using a Leica fluorescence MDR microscope (×100 magnification). Images were captured with a DC500 digital camera microscope using Leica LAS X software at the following exposure times: GFP 0.7 s; Hoechst-33342 0.136 s; bright field 0.62 s (1× gain).

For GFP live imaging in primary human hepatocytes, images of liver-stage parasites were obtained with the Axiovision software (Carl Zeiss) using a Leica DMI4000B fluorescent microscope. Images were analysed with ImageJ software.

**Analysis of mosquito stage development.** For analysis of the mosquito stage, *A. stephensi* mosquitoes were fed with the culture of day 14–17 *P. falciparum* gametocytes using the standard membrane feeding assay (SMFA), as previously described[49]. Before mosquito feeding, the number of exflagellation centres formed in gametocyte cultures was determined. Oocyst numbers in the infected midgut were determined 8–12 days post-infection. To count sporozoites, salivary glands from 20–30 mosquitoes were collected in RPMI-1640 (pH 7.2, Gibco) and homogenised using a grinder. The number of sporozoites from the salivary glands was determined using a Bürker cell counter and by phase-contrast microscopy.

**Analysis of liver stage development.** *P. falciparum* liver stage was cultivated as previously described[55]. Briefly, cryopreserved primary human hepatocytes (Bio-predic International, Saint-Grégoire, France) were thawed and seeded in 384-well plates (Greiner Bio-One, Germany) pre-coated with rat-tail collagen I (BD Bioscience, USA). Human hepatocytes were cultivated at 37 °C in 5% $CO_2$ in William's E medium (Gibco) supplemented with 10% ($v/v$) foetal clone III serum (FCS, Hyclone), 100 u/mL penicillin and 100 μg/mL streptomycin (Gibco), $5 \times 10^{-3}$ g/L human insulin (Sigma-Merck), and $5 \times 10^{-5}$ M hydrocortisone (Upjohn Laboratories SERB, France). After one day, the cells were overlaid with Matrigel (Corning). For infection, Matrigel was removed from hepatocyte culture and sporozoites were added before centrifugation at $560 \times g$ for 10 min at RT and incubation at 37 °C, 5% $CO_2$. Three hours later, infected cultures were covered with Matrigel prior to the addition of fresh medium supplemented with amphotericin B. The medium was renewed every day, until cell fixation with 4% ($w/v$) paraformaldehyde solution. For liver-stage parasites count and size determination, infected cultures were stained with a polyclonal anti-*Plasmodium* HSP70 murine serum revealed with Alexa-Fluor 488-conjugated goat anti-mouse IgG (Invitrogen). DAPI was used to visualize nuclei. Parasite images were obtained with Axiovision software (Carl Zeiss) using a Leica DMI4000B fluorescent microscope and used to determine parasite number and size. Parasite size was determined manually as parasite length in μm using a Cell Insight High Content Screening platform equipped with the Studio HCS software (Thermo Fisher Scientific) in Celis Platform (ICM, La Pitié-Salpêtrière, Paris) as described previously[56].

**NanoLuciferase assay.** For the asexual blood stage, white 384-well plates were used for all NanoLuc assays. For the standard curve, a twofold serial dilution of the GFP-NanoLuc line number was added to a plate at 25 μL/well at 2% haematocrit, and the plate was stored at −20 °C for 24 h for RBC lysis. The following day, plates were thawed at room temperature. Subsequently, NanoLuc assays were performed with the Nano-Glo® Luciferase Assay System (Promega) and Luciferase Cell Culture Lysis 5× Reagent (Promega) according to the manufacturer's instructions, with slight modifications. Briefly, Nano-Glo® Luciferase Assay Substrate was diluted 1:500 with Luciferase Cell Culture Lysis Reagent (1×) and added to the plate at 25 μL/well. The luminescence level (in relative light units; RLU) was measured using the SpectraMax Paradigm microplate reader (Molecular Devices). For the dose-response curve of inhibitors, 100 nL/well of serial dilutions of each compound dissolved in either DMSO, DHA (250 μM dissolved in DMSO; positive control. TCI chemicals), or DMSO alone was dispensed with Echo 550 (Beckman Coulter) on 384-well white plates. GFP-NanoLuc line parasites were synchronised through 5% ($w/v$) D-sorbitol treatment for 10 min and added to the plate at 0.3% parasitaemia and 2% haematocrit at 25 μL/well. The plates were incubated with 5% $O_2$, 5% $CO_2$, and 90% $N_2$ in an anaerobic box at 37 °C for 72 h. The plates were then stored at −20 °C for 24 h for RBC lysis. The NanoLuc assay was performed as previously described.

For NanoLuc assay at the gametocyte stage, GFP-NanoLuc gametocytes were induced as described above. Gametocytes at day 13 (stage V) were used for the standard curve after 50 mM NAG selection (day 4–13). A twofold serial dilution of the gametocyte number was added to a white 96-well plate at 100 μL/well at 2% haematocrit, and the plate was stored at −20 °C for 24 h for RBC lysis. The following day, the plate was thawed at room temperature, and NanoLuc assays were performed as described in the section on asexual blood stage. The assay mix (Nano-Glo® Luciferase Assay Substrate and 1× Luciferase Cell Culture Lysis

Reagent) was added to the plate at 100 μL/well. For the dose–response curve of the inhibitors, day 9 gametocytes (stage III) were used after selection with 50 mM NAG (day 4–9). Serial dilution of each compound dissolved in DMSO (Sigma), epoxomicin (250 μM dissolved in DMSO, positive control, Peptide Institute. Inc), or DMSO alone was dispensed with an E4 XLS electronic multichannel pipette (Mettler Toledo) on a white 96-well plate at 0.4 μL/well. Day 9 gametocytes were diluted to 1% parasitaemia and 2% haematocrit with complete medium containing 50 mM NAG and added to the plate at 100 μL/well. The plates were incubated with 5% $O_2$, 5% $CO_2$, and 90% $N_2$ in an anaerobic box at 37 °C for 72 h. On day 12, the medium in each well was replaced with fresh complete medium without the compounds, and the parasites were incubated for 0, 24, 48, or 72 h. On day 12, 13, 14, or 15, the NanoLuc assay was performed with Intracellular TE Nano-Glo® Substrate/Inhibitor (Promega) according to the manufacturer's instructions (72 h + 0 h, 72 h + 24 h, 72 h + 48 h, 72 h + 72 h protocol in Table 3, NanoLuc inhibitor (+), respectively). Briefly, 0.3 μL of NanoBRET™ Nano-Glo® Substrate, 0.1 μL of Extracellular NanoLuc® Inhibitor, and 50 μL RPMI1640 were mixed and added to the plate at 50 μL/well. The luminescence levels were measured using a SpectraMax Paradigm microplate reader (Molecular Devices). For the 72 h protocol in Table 3, NanoLuc inhibitor (+), the medium change step was skipped and NanoLuc assay was immediately performed with NanoLuc inhibitor on day 12. For the 72 h protocol in Table 3 NanoLuc inhibitor (-), NanoLuc assay was performed without medium change and NanoLuc inhibitor, which is the same method as the NanoLuc assay for the asexual blood stage parasites as described above. For assay miniaturisation with white 384-well plates, 100 nL/well of each compound dissolved in DMSO, epoxomicin (67.5 μM dissolved in DMSO, positive control), or DMSO alone was dispensed with Echo 550 (Beckman Coulter) onto 384-well white plates. Day 12 gametocytes (stage IV) were diluted to 1% parasitaemia and 2% haematocrit with a complete medium containing 50 mM NAG and added to the plates at 25 μL/well. The plates were incubated with 5% $O_2$, 5% $CO_2$, and 90% $N_2$ in an anaerobic box at 37 °C for 72 h. On day 15, NanoLuc assays were performed with intracellular TE Nano-Glo® Substrate/Inhibitor (Promega), as described above. The assay mix (NanoBRET™ Nano-Glo® Substrate, Extracellular NanoLuc® Inhibitor and RPMI1640) was added to the plate at 25 μL/well.

For quantification at the oocyst stage, the *P. falciparum*-infected mosquito midgut was isolated as previously described[49]. Briefly, midguts were dissected from infected mosquitoes 7 days after feeding, and the number of oocysts was determined by microscopy. Each individual midgut was transferred to 100 μL of PBS and frozen until the NanoLuc assay was performed. Salivary gland sporozoites were collected from 20–25 glands and counted (see section above). The collected sporozoites were diluted to appropriate numbers using 100 μL of PBS for the NanoLuc assay in triplicate samples, and mixed with 100 μL of 500-fold diluted Nano-Glo® Luciferase Assay Substrate. Luciferase activity from the infected midguts and sporozoites was measured using a Glomax Multi Detection System Luminometer (Promega) and Instinct software (Promega). Another transfectant, the GFP-NanoLuc reporter line, which we termed Exp227, was used to quantify the mosquito stages (Supplementary Fig. 3).

To investigate the use of luciferase readout for intrahepatic development of the GFP-NanoLuc reporter line, luciferase measurements were compared to the presence of HSP70 positive intrahepatic forms assessed by imaging after the subjection of the cells to atovaquone dose-dependent drug pressure and at different maturation times. To this end, cryopreserved human hepatocytes were seeded in 96-well plates and infected with the transgenic line similar as described earlier[57]. Medium containing different atovaquone concentrations was refreshed during the different maturation periods on a daily basis. For imaging, cells were fixed and stained with anti-HSP70 after 4-, 5- and 6-days post infection as described, and imaged on the ImageXpress PICO (Molecular devices). Images were analysed using CellReporterXpress software.

For the luciferase assay infected hepatocytes were washed once with fresh hepatocyte medium on days 4, 5, and 6 post infection to remove extracellular NanoLuc, followed by the replacement of the medium with a 1:1 mix of hepatocyte medium and Nano-Glo® Luciferase Assay Substrate (Promega). Subsequently, the plates were incubated for 10 minutes on a shaker platform at 250 rpm, 37 °C and the luminescence was determined using a Biotek Synergy 2.

**Cytotoxicity assay using human cells.** The cytotoxicity of hit compound OU0074008 was assessed against the HepG2 human cell line (American Type Culture Collection). HepG2 cells in DMEM (Gibco) with 10% ($v/v$) foetal bovine serum (Sigma) were seeded on a clear 96-well plate at $1 \times 10^4$ cells/well and incubated at 37 °C under 5% $CO_2$ (day 0). On day 1, the medium was replaced with fresh medium, and 0.4 μL/well of either OU0074008 serially diluted by dissolving in DMSO or DMSO alone was dispensed with an E4 XLS electronic multichannel pipette. After 72 h of incubation at 37 °C under 5% $CO_2$ (day 4), the viability of HepG2 cells was measured using a Cell Counting kit-8 (Wako) according to the manufacturer's instructions. Absorbance was measured using a SpectraMax Paradigm microplate reader (Molecular Devices) at 450 nm.

**Screening of Osaka University chemical library.** For the asexual blood-stage, 1,920 compounds were screened in 384-well plates by applying the same assay conditions as for the duplicates described above (see the section "NanoLuciferase assay, Asexual blood stage"). For gametocyte-stage screening, 30 compounds

effective against the asexual blood-stage were tested in 96-well plates in duplicate, as described above (see the section "NanoLuciferase assay, gametocyte stage"). The additional incubation step after medium change was skipped, and the NanoLuc assay was performed with an extracellular NanoLuc inhibitor immediately after the medium change (72 h + 0 h protocol). The IC$_{50}$ values of the hit compound OU0074008 against the asexual blood and gametocyte stages were determined using the same assay methods with serial dilution of the compound. The inhibitory activity of the hit compound OU0074008 against the liver stage was examined using a NanoLuc expressing reporter line previously established[28]. The compound OU0074008 was previously synthesised and described[47].

### Counter assays

*Lactate dehydrogenase (LDH) assay.* The anti-malarial activity of the hit compound (OU0074008) was confirmed by the LDH assay as an alternative method. The LDH assay was performed as previously described[58] The GFP-NanoLuc reporter line was incubated with OU0074008 or DMSO control in 96-well plate at 37°C for 72 hours with 0.3% initial parasitemia. After 72 h of incubation, parasite growth was determined by diaphorase-coupled LDH assay. The absorbance of each well was measured at 655 nm using SpectraMax Paradigm® Multi-Mode microplate reader (Molecular Devices, San Jose, CA, USA). The inhibition rate was calculated with the absorbance of 1 μM DHA wells defined as 100% inhibition.

*NanoLuc inhibition assay.* The inhibitory effect of the hit compound (OU0074008) was evaluated as previously described[46]. Briefly, the whole lysate of the GFP-NanoLuc reporter line from the asexual blood or gametocyte stage was prepared by a freezing-thawing process of the in vitro culture of the transgenic lines. Subsequently, whole lysates were incubated with OU0074008, Intracellular TE Nano-Glo® Substrate/Inhibitor (Promega, positive control), or DMSO (negative control) at 37 °C for 10 min. RLU from each sample was measured as described above.

### Statistics and reproducibility

The quality of the screening system was evaluated by calculating statistical parameters (Z'-factor, S/N, S/B, CV$_{min}$, and CV$_{max}$), as previously reported[59]. The formula of each parameter is as follows: Z'factor = 1−(3 × SD$_{100\%}$ + 3 × SD$_{0\%}$)/(Av$_{100\%}$−Av$_{0\%}$), S/B = Av$_{100\%}$/Av$_{0\%}$, S/N = (Av$_{100\%}$-Av$_{0\%}$)/ SD$_{0\%}$, %CV$_{max}$ = SD$_{100\%}$/Av$_{100\%}$ × 100, %CV$_{min}$ = SD$_{0\%}$/Av$_{0\%}$ × 100. Here, SD$_{100\%}$ and SD$_{0\%}$ mean the standard deviation of signal intensity in negative control wells and positive control wells, and Av$_{100\%}$ and Av$_{0\%}$ mean the mean of signal intensity in negative control wells and positive control wells, respectively. DMSO wells were defined as negative control wells, and 1 μM DHA wells (for asexual blood-stage) or 1 μM or 270 nM epoxomicin wells (for gametocyte-stage) were defined as positive control wells. The IC$_{50}$ value of each test compound were determined in triplicate or quadruplicate, and calculated using GraphPad Prism 9.0 software (GraphPad Software Inc., San Diego, CA, USA). The detail of the replicates are described in the relevant Figure legends. Duplicates to eight replicates were used in this study.

### Reporting summary

Further information on research design is available in the Nature Portfolio Reporting Summary linked to this article.

## Data availability

All datasets generated for this study are included in the article/Supplementary Material. Original blots/gels are included in Supplementary Fig. 7. Source data underlying the figures are available in Supplementary Data. All other data are available from the corresponding authors (or other sources, as applicable) on reasonable request.

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

## Acknowledgements

We are grateful to Catherin Marin-Mogollon and Chris J. Janse at Leiden University for valuable discussions, and Minako Yoshida, Megumi Tanaka, Takaya Sakura, Ryuta Ishii, and Fuyuki Tokumasu for technical assistance and helpful discussions. This paper is dedicated to the memory of our friend and colleague, Dr. Shahid Khan, who passed away on the 4th of October, 2019. We would like to acknowledge the Radboud Technology Center Microscopy of Radboud University Medical Center for providing access to their facilities. We are very grateful to Mallaury Bordessoulles and Nadia Amanzougaghene from CIMI-Paris for their help with the culture of *P. falciparum* liver-stage in primary human hepatocytes. We also thank the Nagasaki Red Cross Blood Center for providing human RBC and plasma samples. We would like to thank Editage (www.editage.com) for English language editing. SM was partially supported by the European Union's Horizon 2020 Research and Innovation Program under grant agreement No. 733273. This work was partially supported by SHIONOGI & Co. Ltd. This research was partially supported by the Platform Project for Supporting Drug Discovery and Life Science Research (Basis for Supporting Innovative Drug Discovery and Life Science Research (BINDS)) from AMED under grant number JP21am0101001(support number 3361). This work was supported in part by JSPS KAKENHI (Grant Numbers 20K22767, 21K06994 and 22K15452) and the Japan Agency for Medical Research and Development (AMED) (grant numbers 22jk0210036h0002 and 22wm0325051h0001). This study was supported by the Ohyama Health Foundation for S.M. The funders had no role in the study design, data collection and analysis, decision to publish, or manuscript preparation.

## Author contributions

Y.M. and S.M. came up with the study concept and design. Y.M., M.V., F.G., P.B., H.K., V.S., and S.M. acquired the data. Y.M., M.V., V.S., M.A., D.K.-I., K.D., B.F.-F., and S.M. conducted analysis and interpretation of the data. Y.M., M.V., F.G., H.K., M.A., V.S., and S.M. wrote the draft of the manuscript. Y.M., M.V., V.S., D.K.-I., K.D., B.F.-F., and S.M. critically revised the manuscript. S.Y. and M.A. provided technical and/or material support. Y.M. and S.M. supervised the study. All authors reviewed the manuscript.

## Competing interests

The authors declare the following competing interests: Y.M., D.K.-I. and S.M. have the following financial relationships to disclose with the grant/research funding from SHIONOGI & CO., LTD. M.V. and K.D. was employed by TropIQ Health Sciences. The remaining authors declare that the research was conducted in the absence of any commercial or financial relationships that could be construed as potential conflicts of interest.

## Ethics statement

Primary human liver cells were purchased from Biopredic International (Saint-Grégoire, France), Lonza Bioscience, or Tebu-Bio.

## Additional information

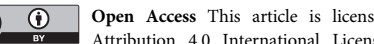

