## [Peer Review File · Communications Biology]

Reviewers' comments:

Reviewer #1 (Remarks to the Author):

Miyazaki Y. and colleagues have generated a *Plasmodium falciparum* reporter line that expresses GFP and NanoLuc. NanoLuc expression has been confirmed and employed in the evaluation of the anti-plasmodial activity of compounds against asexual blood, gametocyte and liver parasite stages. Employment of this transgenic parasite for the screening of a library of 1920 compounds led to the identification of a hit compound that is active against both asexual blood and gametocyte *P. falciparum* stages. Even though other reporter parasite lines that express luciferase (firefly or NanoLuc) have been described, this is the first *P. falciparum* line expressing NanoLuc that has been validated for the evaluation of anti-plasmodial activity against gametocyte and liver stage parasites, besides asexual blood stage parasites. The greater brightness of the NanoLuc signal, when compared to firefly luciferase, increases the sensitiveness of parasite detection. Furthermore, the confirmation of its expression across the parasite's life cycle makes it a useful tool in the search for compounds with multistage activity. Overall, the paper is clearly written and the claims are supported by the evidence presented. However, I have some minor concerns that the authors should address before publication of this manuscript.

MANUSCRIPT TEXT

Line 48 – “Parasites in the hepatocytes eventually lead to rupture”. Intra-hepatic parasites are released from hepatocytes in merosomes, not upon hepatocyte rupture. The sentence should be rephrased accordingly.

For clarity, “asexual stages” could instead be referred to as “asexual blood stages” across the manuscript text.

Line 131 – “For subsequent analysis, we mainly used the GFP-NanoLuc reporter line”. This sentence is too vague, and the authors should clarify when and why one or the other parasite line was used. The mCherry reporter line is not clonal. If this was a reason for not using it, this should be stated.

Line 185 – please consider revising the statement as it appears that the results of this section support that the parasite line is suitable for asexual blood stage assays, a claim that in fact is supported by the findings of the previous results section.

Line 213 (liver stage section) and Figure 5:

- a direct correlation between infection and bioluminescence, like the ones carried out for the remaining parasite stages, is missing. RLU could be correlated with parasite development through time, as in reference 14. Alternatively, RLU could be correlated with infection quantification by qRT-PCR, as in Ploemen IHJ 2009 (doi:10.1371/journal.pone.0007881).
- The live cell imaging presented in Fig 5A only shows a green signal and, in Fig 5B and 5C, intra-hepatic parasite numbers and areas were assessed through anti-HSP70 staining and DAPI labeling. Have the authors ever performed a double staining against GFP and a parasite protein, such as HSP70, to show colocalization of both?

Line 239 - Since the authors have established and validated the gametocyte assay in a 384w format, was there a reason for this not being employed in the HTS of the compounds from the Osaka University chemical library? This should be briefly clarified in the manuscript text.

Lines 243-245 and Figure 6G: The IC₅₀ value against HepG2 cells is said to be higher than that against the asexual blood stage parasite. However, this value is not presented and the data does not appear to allow for an IC₅₀ determination. The axis labeling is also misleading, as it does not clearly indicate that what is being measured is toxicity towards the human hepatic cell line at the compound concentrations tested. This is confusing and should be clarified.

Line 299 – This sentence could be rephrased for clarity. As it is written, it appears that the activity

of ATQ was evaluated with a higher sensitivity than the already published parasite line of reference 14. However, the IC50 values obtained in both systems are within the same order of magnitude and there is no way to directly correlate one to the other.

Lines 408-410 – It is said that “Exp227 was used for the NanoLuc reaction at oocyst and sporozoite stages”. Has this line been characterized? This data should be included as supplementary results.

The authors could consider briefly mentioning in the discussion other, non-human, reporter Plasmodium parasite lines expressing luciferase that allow, for example, to test the activity of compounds in vivo, in rodent malaria models. In this line of thought, although not mandatory for publication, demonstrating that the GFP-NanoLuc reporter parasite line could be used to measure infection through bioluminescence in the liver or blood of liver- or blood-humanized mice would certainly be of added value to the manuscript.

TABLES, FIGURES, AND FIGURE LEGENDS

Figure 1D – it is not clear in which sample this expression is being analysed. Please clarify.

Figure 2D – “Dose-response curves of established antimalarial compounds” would be a more suitable description of the plots, instead of “Determination of IC50 value”, given that this is only shown in the table. Inhibition is represented as a % of the control, however, it is not clear what that control is.

Figure 3D – the control to which inhibition is being normalized to should be described in the figure legend.

Figure 4 – The letters are misaligned and Fig 4C images could benefit from a size increase.

Figure 5 – figure legend for D and E mentions a dose-dependent inhibition of intrahepatic parasite development. However, figure 5E shows the impact of ATQ on the percentage of infected hepatocytes, which is not a measure of intra-hepatic parasite development. Have the authors assessed EEF areas? RLU measurements will not allow for the discrimination between the impact of a drug on hepatocyte invasion and intra-hepatic development. A direct comparison between the data of Fig 5D and Fig 5E should not be made without discussing these points.

Figure 6 – Inhibition is represented as a % of the control, however, it is not clear what that control is. In Fig 6G, the axis is misleading and it does not immediately convey the idea that the readout is cytotoxicity.

Table 1 – the SD of stage III gametocytemia for GFP-NanoLuc should be within brackets for consistency. The meaning of (4 exp.), etc, should be clarified. Given that stage III gametocytemia was not determined for Pf NF54 WT but is referred to in the text as being normal, there should be a literature reference to support this conclusion.

Table 2 – It should be indicated what is represented after the +/- of the IC50. The table description should indicate the number of biological replicates from which the IC50 value is being extrapolated.

Figure S1 – there is a contradiction between the expected amplicon size shown in S1A resulting from the use of primers p1 and p5 (1.5 kb) and that shown in S1B and described in the figure legend (1.1 kb). This should be clarified. Do the authors have an explanation for the presence of a band of ~1.5 kb in the first gel lane?

For all correlation graphs, it is said in the figure legend that the solid line indicates the mean RLU from triplicate samples. Isn't that instead indicated by the dots and doesn't the line correspond to the correlation itself?

Reviewer #2 (Remarks to the Author):

The study by Miyazaki et al demonstrates the usefulness of transgenic parasites for drug development in malaria. The transgenic reporter line that was produced in this study overcomes several limitations with previously generated reporter lines. These include that; 1) it is a dual reporter for fluorescent read out and luciferase-based readout, 2) the nanoluciferase used is significantly brighter than other luciferases that have been used, which is a significant advantage when using low cell numbers such as for gametocytes, mosquito stages and liver stages, and 3) the reporter is introduced into the genome in a stable and marker free method making it suitable for all parasite lifecycle stages. Furthermore, the authors made improvements to assay format that overcome limitations with leftover signal from dead parasites in the gametocyte assay. Using these methods, the authors demonstrated that the reporter line is suitable for drug screening throughout the lifecycle stages. As a proof of concept, the authors performed a HTS screen of 1920 compounds against asexual stages, and 30 hits from the screen against gametocyte stages. This led to the finding of OU0074008 with activity against both stages that could be a promising lead compound for further development. However, several issues do need to be addressed. The main issue is the lack of clear reporting of the number and type of replicates used in each figure (see below).

Major comments:

1. In the majority of figures there is insufficient detail on the number of replicates shown. E.g. 2B and 2C have no information about replicates. For figure 5, it states the mean of quadruplicates. It is unclear if these are technical duplicates from a single experiment, or biological quadruplicates, in which case was it a single technical replicate in each condition? These comments apply to the majority of figures in this study.
2. Lines 241-242. A common problem with luciferase as the readout for assays is the ability of small molecules to directly inhibit luciferase activity. Given the significant difference between the asexual and gametocyte assay, it is very likely it is specific against parasite growth in asexuals. However, it may be that the gametocyte activity is due to luciferase inhibition. Either of the following experiments would clarify this. 1) A counter screen for luciferase inhibition. Directly test for inhibition of luciferase using lysate from asexual parasites (lysing them and adding drug – at the same concentrations used in the gametocyte assay - directly to the lysate for a short period prior to adding substrate). 2) repeat the gametocyte assay using an alternative read out for the assay.

Minor comments

3. Figure 2B, how was the serial dilution performed? If it was diluted into the equivalent amount of blood lysate the signal at low parasitemia becomes quenched more so (due to hemoglobin) than if the sample was diluted in PBS or equivalent (Azevedo 2014).
4. A comment on the advantages and limitations of the gametocyte assay would be useful. For example, do you know if your assay is suitable for separating out early vs late stage gametocyte activity (e.g. Duffy et al, 2013, Malaria Journal).
5. Figure 4D. It is very difficult to see the individual data points at the low end. A log scale, or a split axis would help this.
6. Figure 5C, the labels are positioned in such a way that they look like two data points in D7.

Reviewer #3 (Remarks to the Author):

Miyazaki et al. present a versatile *Plasmodium falciparum* dual reporter line based on integrated

bright Nanoluc luciferase and the fluorescent GFP (or mCherry). The line does not contain a selectable marker and shows reporter expression throughout the life cycle. The others nicely show that the constitutively expressed reporters do not appear to have any effect on the growth characteristics of the parasites at the different developmental stages of the life cycle. They propose that this line is suitable for the multi-stage evaluation of anti-malarial drug efficacy and using this line in HTS against asexual blood stage parasites and gametocytes, they identify OU0074008 as a novel anti-malarial compound.

While many different *P. falciparum* reporter lines have been developed, this new reporter line is attractive, because it is marker-less, has constitutive expression of GFP for visualization/isolation and the bright Nanoluc for bioluminescence read-out for drug assays at multiple stages of the parasite life cycle and appears to have a normal viability. This 'one-for-all' line could simplify screening of compounds directed against different parts of the life cycle of *P. falciparum*.

This manuscript represents a nice piece of work with investigations covering the full life cycle of the parasite. In my opinion however, some of the claims, especially claims regarding the performance of the parasite line during the liver stage part of the life cycle are not fully substantiated (see below).

Major comments

1. The analyses of the liver stage parasites by bioluminescence measurements appear to be limited. In the abstract (line 33) 'a quantitative' NanoLuc signal in liver stages (amongst other stages) is mentioned and it is stated that an assay system was established for asexual, gametocyte, and liver stages (line 35). In line 276 it is stated that luminescence and parasite number are well correlated, but this is not (clearly) shown for liver stages. And in the conclusion (lines 322 and 323) it is stated that a robust drug assay protocol is established for asexual-, gametocyte-, and liver-stage parasites. For asexual stages and gametocytes, relationship between parasite number and RLU, drug assay parameters (Z' factors etc.) have been described and multiple drugs were tested. However, for liver stage parasites, this is not the case. Only data of a single (?) drug experiment with atovaquone are shown. To substantiate the claim that a robust liver stage drug assay based on bioluminescence with these parasites has been developed, more experimentation is needed. Similar to the work performed on the asexual- and gametocyte stages, the relationship between RLU values and parasite numbers needs to be assessed and multiple drug assays to determine activity with different drugs should be performed for liver stages as well.

2. Also, the claim that the activity of atovaquone against liver stage *P. falciparum* can be evaluated with a 'higher sensitivity' (line 301) compared to a previously established similar system by Marin-Mogollon et al (ref 14) needs more extensive experimentation (in fact IC50 values from both methods appear to be similar).

3. In the light of using a single reporter line for multiple stages of the life cycle, it would make sense to test the novel identified anti-malarial compound OU0074008 for asexual and gametocyte stages also in the liver stage assay.

Minor points

1. The use of a single reporter line for the investigation of multiple stages of the life cycle would simplify the drug screening work. However, aren't the authors concerned that cross-contamination of stages (i.e. presence of remaining blood stages in the gametocyte preparation) could give problems in assessing drug activity against specific stages?

2. For the figures with IC50 curves, the following details should be clarified and mentioned: how many times was the assay performed (independently); how many wells were used per assay (96 or 384 well format); what is depicted, for example triplicates of wells or triplicates of assays?

3. The authors mention also the development of a mCherry-NanoLuc line. They do show some characteristics of this line as well. While this is good to know, the line does not play any role in the main message of the manuscript. To keep the main text focused on the GFP line, the authors may consider to put all mCherry-NanoLuc data in one figure in the supplement and to just note that similar to the GFP-line, an mCherry line was developed with similar characteristics.

4. The rationale for using NanoLuc is not completely clear from the introduction (also, please provide a ref for NanoLuc in line 87).
5. Line 112/113: was the in vitro growth of the mCherry-NanoLuc line also comparable to the WT strain?
6. Lines 166, 167: wouldn't these prolonged incubation times affect parasite viability? Did the authors check whether these gametocytes could still be transmitted? And what is the transmission rate of the gametocytes after the different incubation times?
7. Line 179: could the need for the NanoLuc inhibitor indicate prolonged stability of the protein? If so, do the authors think that it would be an option to develop a NanoLuc reporter with a degradation domain to counteract this issue?
8. Lines 225, 226: were all HSP70 parasites also GFP positive (indication of the stability of the transgenes in the genome)?
9. Figure 5: panel E: on the y-axis % infected hepatocytes is stated. How was this calculated? What is the number of liver stages per well? Is this a single assay from which means and SD of triplicate wells are shown? I would strongly suggest using a more commonly used parameter, such as % of untreated control instead.
10. Lines 299-302: the authors state that the liver stages could be measured with high sensitivity. I do think that more data are needed for this statement (see above). Also, the RLU values appear to be low (around 4000) if these reflect parasite numbers around 50-100 per well. In papers describing Nanoluc in liver stage parasites from other malaria parasites (de Niz et al, Malar J. 2016;15:232 and Voorberg-van der Wel et al, Anal Chem 202, 92, 9, 6667-6675) RLU values are much higher. Can the authors comment? How long did the authors measure per well? Did the authors deviate from the protocol of the manufacturer (and/or the other papers)?
11. Figure 2, panel B: the RLU values of low numbers of iRBC are below the background value of uninfected RBC. Do the authors have any idea why this is the case?
12. M&M, lines 394, 395: can the authors give a few more details here? Was this a mixed population of parasites, or for example, purified schizonts? What was the starting parasitemia (and much time did it take until parasitemia reached 5%)?
13. M&M, lines 487-490: why did the authors use Matrigel one day after hepatocyte seeding and after parasite inoculation? Didn't the removal of Matrigel affect the hepatocyte monolayer?
14. M&M, lines 497: was the counting performed manually or did the authors use a script?
15. M&M, lines 518-520: what was the end parasitemia of the assay?

Response letter for the manuscript entitled "A versatile *Plasmodium falciparum* reporter line expressing NanoLuc enables highly sensitive multi-stage drug assays"

Our responses are indicated in a blue font.

Reviewers' comments:

We sincerely thank the Reviewers and Editors for providing their valuable comments and suggestions to improve our manuscript. In response to the issues raised by the reviewers, we performed several new experiments and thoroughly revised the manuscript to address all the comments raised. We believe that our new data and revisions have greatly improved the quality of the manuscript, thereby making it suitable for publication.

Reviewer #1 (Remarks to the Author):

Miyazaki Y. and colleagues have generated a *Plasmodium falciparum* reporter line that expresses GFP and NanoLuc. NanoLuc expression has been confirmed and employed in the evaluation of the anti-plasmodial activity of compounds against asexual blood, gametocyte and liver parasite stages. Employment of this transgenic parasite for the screening of a library of 1920 compounds led to the identification of a hit compound that is active against both asexual blood and gametocyte *P. falciparum* stages. Even though other reporter parasite lines that express luciferase (firefly or NanoLuc) have been described, this is the first *P. falciparum* line expressing NanoLuc that has been validated for the evaluation of anti-plasmodial activity against gametocyte and liver stage parasites, besides asexual blood stage parasites. The greater brightness of the NanoLuc signal, when compared to firefly luciferase, increases the sensitiveness of parasite detection. Furthermore, the confirmation of its expression across the parasite's life cycle makes it a useful tool in the search for compounds with multistage activity. Overall, the paper is clearly written and the claims are supported by the evidence presented. However, I have some minor concerns that the authors should address before publication of this manuscript.

We would like to thank the reviewer for the positive comments and informative suggestions on how to improve the manuscript.

MANUSCRIPT TEXT

Line 48 – “Parasites in the hepatocytes eventually lead to rupture”. Intra-hepatic parasites are

released from hepatocytes in merosomes, not upon hepatocyte rupture. The sentence should be rephrased accordingly.

As per the reviewer's suggestion, we have revised the sentence as follows:

Line 48-50 "Parasites in the hepatocytes are eventually released into the bloodstream, and subsequently invade RBCs for further proliferation."

For clarity, "asexual stages" could instead be referred to as "asexual blood stages" across the manuscript text.

We have revised all the instances of "asexual stages" to "asexual blood stages" in the manuscript, as suggested.

Line 131 – "For subsequent analysis, we mainly used the GFP-NanoLuc reporter line". This sentence is too vague, and the authors should clarify when and why one or the other parasite line was used. The mCherry reporter line is not clonal. If this was a reason for not using it, this should be stated.

In addition to the reasons for the utility of GFP fluorescence for FACS sorting, the clonality of the transgenic line constitutes one of the important reasons. We revised the sentence to clarify this as follows: (Line 135-140)

"Subsequently, we successfully obtained a clonal line of the GFP-NanoLuc reporter line, but not the mCherry-NanoLuc line. For downstream analysis, we mainly used the GFP-NanoLuc reporter line owing to the clonality of an isolated clonal line (cl.1) and the utility of GFP for sorting specific life cycles, as exemplified by several studies wherein different *Plasmodium* species expressing GFP were employed."

Line 185 – please consider revising the statement as it appears that the results of this section support that the parasite line is suitable for asexual blood stage assays, a claim that in fact is supported by the findings of the previous results section.

As suggested, we have revised this sentence to clarify the meaning of this paragraph. We have also deleted "as well as for assays performed at the asexual stage." from the original sentence (Line 190-191).

"These findings demonstrated that the GFP-NanoLuc line is applicable for *P. falciparum* gametocytocidal assays."

Line 213 (liver stage section) and Figure 5:

– a direct correlation between infection and bioluminescence, like the ones carried out for the remaining parasite stages, is missing. RLU could be correlated with parasite development through

time, as in reference 14. Alternatively, RLU could be correlated with infection quantification by qRT-PCR, as in Ploemen IHJ 2009 (doi:10.1371/journal.pone.0007881).

To address this comment, we have now included analyses of parasite numbers versus luciferase signals for day 4, 5 and 6 post infection in revised Figure 5H (Line 792-793). Revised Figure 5 is shown below.

The corresponding sentence related to Fig.5H has now been included in the Result section (Line 240-242).

“Overall, we observed a high degree of correlation between luminescence values and number of liver stage schizonts (Fig. 5H; $R^2 = 0.90; 0.95; 0.96$ for day 4, 5 and 6, respectively).”

- The live cell imaging presented in Fig 5A only shows a green signal and, in Fig 5B and 5C, intra-hepatic parasite numbers and areas were assessed through anti-HSP70 staining and DAPI labeling. Have the authors ever performed a double staining against GFP and a parasite protein, such as HSP70, to show colocalization of both?

We did not perform double staining against GFP and a parasite protein such as HSP70. Staining for HSP70 and DAPI was done in order to quantify parasites' number and size on fixed cells accurately since fixation affects the intensity of the GFP fluorescence signal.

Line 239 - Since the authors have established and validated the gametocyte assay in a 384w format, was there a reason for this not being employed in the HTS of the compounds from the Osaka University chemical library? This should be briefly clarified in the manuscript text.

In this study, we screened 1920 compounds using asexual blood stages and selected only 30

compounds for the downstream gametocytocidal assay. As shown in Table 3, the 96-well plate gametocytocidal assay is a more robust drug assay system than the 384-well format, although the throughput of the 96-well format is lower. To screen only 30 compounds, we considered the 96-well formats as a suitable assay system.

We have revised the sentence in Lines 255-258 to address this issue as follows

“For the gametocytocidal assays, we applied the 72 h+0 h protocol with the extracellular NanoLuc inhibitor in a 96-well plate format considering that it exerted high assay quality and required a short time (Table 3).”

Lines 243-245 and Figure 6G: The IC₅₀ value against HepG2 cells is said to be higher than that against the asexual blood stage parasite. However, this value is not presented and the data does not appear to allow for an IC₅₀ determination. The axis labeling is also misleading, as it does not clearly indicate that what is being measured is toxicity towards the human hepatic cell line at the compound concentrations tested. This is confusing and should be clarified.

We agree with the reviewer’s comment. To address this, we repeated the cytotoxicity assay by applying a more dynamic range of the compound, which determined the IC₅₀ value of OU0074008 against HepG2 cells. This new data has been added to Fig. 6G by changing the Y-axis to cytotoxicity (% of control), as suggested by the reviewer. The new Fig.6 is shown below.

Line 299 – This sentence could be rephrased for clarity. As it is written, it appears that the activity of ATQ was evaluated with a higher sensitivity than the already published parasite line of

reference 14. However, the IC50 values obtained in both systems are within the same order of magnitude and there is no way to directly correlate one to the other.

We agree with this suggestion. We have rephrased this sentence as follows (Line 330-334):

‘Moreover, we demonstrated that the activity of atovaquone and MMV390048 against liver stage *P. falciparum* in primary human hepatocytes can be evaluated using our drug assay system with brighter luminescence when compared to a previously established system using Firefly luciferase.

Lines 408-410 – It is said that “Exp227 was used for the NanoLuc reaction at oocyst and sporozoite stages”. Has this line been characterized? This data should be included as supplementary results.

As per the reviewer’s suggestion, we have included new Figure S3 that shows the characteristics of the GFP-NanoLuc reporter line Exp227 line and related sentences to the Method section (Line 444-445). Figure S3 is shown below.

The authors could consider briefly mentioning in the discussion other, non-human, reporter Plasmodium parasite lines expressing luciferase that allow, for example, to test the activity of compounds in vivo, in rodent malaria models. In this line of thought, although not mandatory for

publication, demonstrating that the GFP-NanoLuc reporter parasite line could be used to measure infection through bioluminescence in the liver or blood of liver- or blood-humanized mice would certainly be of added value to the manuscript.

As suggested, we have added new sentences in the Discussion to revise this section (Line 304-309).

“Moreover, immunocompromised mice engrafted with human red blood cells and primary human hepatocytes were established and is now an instrumental resource for preclinical drug and vaccine safety and efficacy screens. For *in vivo* studies using humanized mice, the GFP-NanoLuc reporter line generated in this study may be a valuable tool to investigate how the parasite grows and evaluate anti-malarial compounds in the infection model.”

TABLES, FIGURES, AND FIGURE LEGENDS

Figure 1D – it is not clear in which sample this expression is being analysed. Please clarify.

A mixed stage of the parasite was used for Western blotting. We have added this information to the Figure legend and the corresponding Methods section (Line 459, 716).

Figure 2D – “Dose-response curves of established antimalarial compounds” would be a more suitable description of the plots, instead of “Determination of IC50 value”, given that this is only shown in the table. Inhibition is represented as a % of the control, however, it is not clear what that control is.

As suggested, we have revised the legend for Fig.2D to ‘Dose-response curves of established antimalarial compounds’ (Line 737-738). The percentage of inhibition was calculated by dividing RLU (with compound) by RLU (without compound, DMSO control) and subsequent 100-fold multiplications. To describe the control, we have added an additional description in the legend for Fig.2 and Fig.3 (Line 738-740, 766-769).

“Dose-response curves of established antimalarial compounds. The GFP-NanoLuc reporter lines were cultured in different concentrations of indicated anti-malarial compounds, DMSO (negative control well) or 1 μ M DHA (positive control well).”

“Dose-response curves of epoxomicin and atovaquone for gametocytes. The GFP-NanoLuc reporter lines were cultured in different concentrations of the indicated anti-malarial compounds, DMSO (negative control well) or 1 μ M epoxomicin (positive control well).

Figure 3D – the control to which inhibition is being normalized to should be described in the figure legend.

This issue is the same as the one mentioned above. The legend for Fig.3 has been revised as

according to the above-mentioned modification (Line 766-769).

Figure 4 – The letters are misaligned and Fig 4C images could benefit from a size increase. We have revised Fig.4C based on this suggestion. New Fig.4 is shown below.

Figure 5 – figure legend for D and E mentions a dose-dependent inhibition of intrahepatic parasite development. However, figure 5E shows the impact of ATQ on the percentage of infected hepatocytes, which is not a measure of intra-hepatic parasite development. Have the authors assessed EEF areas? RLU measurements will not allow for the discrimination between the impact of a drug on hepatocyte invasion and intra-hepatic development. A direct comparison between the data of Fig 5D and Fig 5E should not be made without discussing these points.

Indeed, the luciferase assay does not discriminate between invasion and parasite growth. We have clarified this in the text (Line 228-244) and revised Figure 5 (See above). It now shows the effect of atovaquone on luciferase signals and the absolute number of parasites per well. In addition, we show the effect on the total parasite area, which is the sum of the number of parasites and their sizes. IC₅₀ values of atovaquone calculated from each dose-response curve have been included in Table 2.

Figure 6 – Inhibition is represented as a % of the control, however, it is not clear what that control is. In Fig 6G, the axis is misleading and it does not immediately convey the idea that the readout is cytotoxicity.

This issue is the same as that described above. The legend for Figure.6G has been revised as

according to the above-mentioned modification (Line 834-836).

“The cells were exposed for 72 h with indicated concentration of OU0074008 or DMSO (control well) on a 96 well plate and cytotoxicity assay using CCK-8 was performed.”

Table 1 – the SD of stage III gametocytemia for GFP-NanoLuc should be within brackets for consistency. The meaning of (4 exp.), etc, should be clarified. Given that stage III gametocytemia was not determined for Pf NF54 WT but is referred to in the text as being normal, there should be a literature reference to support this conclusion.

Table 1 has been revised to address this issue. We have now included data on stage III gametocytemia with the mean and SD values (Line 839). In addition, we have added a description of ‘exp’ into Table 1 and 3 (Line 843, 862).

Table 2 – It should be indicated what is represented after the +/- of the IC50. The table description should indicate the number of biological replicates from which the IC50 value is being extrapolated.

We have revised Table 2 according to your suggestion (Line 847-848).

Figure S1 – there is a contradiction between the expected amplicon size shown in S1A resulting from the use of primers p1 and p5 (1.5 kb) and that shown in S1B and described in the figure legend (1.1 kb). This should be clarified. Do the authors have an explanation for the presence of a band of ~1.5 kb in the first gel lane?

Thank you for spotting this error, which shows that the amplicon by 5'-int primers was 1.5 kb. We have revised this information based on your suggestion. We rarely amplify a ~1.5 kb band in WT by primer P1/P5, as we previously reported (Fig.4B in Miyazaki S et al. Front Cell Infect Microbiol, PMID PMC7298075). We recognize that this 1.5 kb band is non-specific and does not reflect the integration of the donor plasmid. To clarify this confusing data, we have added the following sentence to the legend of Fig.S1 (Line 881-882).

“The weak 1.5 kb band with the 5'-int-primers is a non-specific fragment which is only present in the WT and not in the transgenic line.”

For all correlation graphs, it is said in the figure legend that the solid line indicates the mean RLU from triplicate samples. Isn't that instead indicated by the dots and doesn't the line correspond to the correlation itself?

Thank you for spotting this error. As suggested, the dots indicate the mean value, and the solid line indicates linear regression. We have revised this information in all of the figure legends (Line 732-734, 757-759, 788-790).

Reviewer #2 (Remarks to the Author):

The study by Miyazaki et al demonstrates the usefulness of transgenic parasites for drug development in malaria. The transgenic reporter line that was produced in this study overcomes several limitations with previously generated reporter lines. These include that; 1) it is a dual reporter for fluorescent read out and luciferase-based readout, 2) the nanoluciferase used is significantly brighter than other luciferases that have been used, which is a significant advantage when using low cell numbers such as for gametocytes, mosquito stages and liver stages, and 3) the reporter is introduced into the genome in a stable and marker free method making it suitable for all parasite lifecycle stages. Furthermore, the authors made improvements to assay format that overcome limitations with leftover signal from dead parasites in the gametocyte assay. Using these methods, the authors demonstrated that the reporter line is suitable for drug screening throughout the lifecycle stages. As a proof of concept, the authors performed a HTS screen of 1920 compounds against asexual stages, and 30 hits from the screen against gametocyte stages. This led to the finding of OU0074008 with activity against both stages that could be a promising lead compound for further development. However, several issues do need to be addressed. The main issue is the lack of clear reporting of the number and type of replicates used in each figure (see below).

We would like to thank the reviewer for the useful suggestions for enhancing the quality of our manuscript that we have incorporated in the updated manuscript.

Major comments:

1. In the majority of figures there is insufficient detail on the number of replicates shown. E.g. 2B and 2C have no information about replicates. For figure 5, it states the mean of quadruplicates. It is unclear if these are technical duplicates from a single experiment, or biological quadruplicates, in which case was it a single technical replicate in each condition? These comments apply to the majority of figures in this study.

We completely agree with this opinion and have included biological and technical replicates in the Figure legends (Line 714, 732, 740-741, 757-758, 769, 788-789, 798-799, 801-802, 804-805, 809, 811, 836-838).

2. Lines 241-242. A common problem with luciferase as the readout for assays is the ability of small molecules to directly inhibit luciferase activity. Given the significant difference between the asexual and gametocyte assay, it is very likely it is specific against parasite growth in asexuals.

However, it may be that the gametocyte activity is due to luciferase inhibition. Either of the following experiments would clarify this. 1) A counter screen for luciferase inhibition. Directly test for inhibition of luciferase using lysate from asexual parasites (lysing them and adding drug – at the same concentrations used in the gametocyte assay - directly to the lysate for a short period prior to adding substrate). 2) repeat the gametocyte assay using an alternative read out for the assay.

We agree with this suggestion. To address these issues, we performed new experiments regarding the direct inhibition of NanoLuc by OU0074008 using a lysate of asexual blood and gametocyte stages (Counter assay). In addition to the NanoLuc inhibition assay, we performed an LDH assay using asexual blood stages treated with OU0074008 to further verify the antimalarial activity of OU0074008. These data are shown in Fig.S5 (Line 944, and See below), and the corresponding sentences below have been included in the Results and Method sections (Line 263-268, Line 659-678).

“To verify the anti-malarial activity of OU0074008, we performed a lactate dehydrogenase (LDH) assay and confirmed that the IC₅₀ value from different types of viability assay was equivalent to that from the NanoLuc assay (Fig.S5). Furthermore, the NanoLuc inhibition assay with OU0074008 suggests that there is no obvious NanoLuc inhibitory activity of the hit compound (Fig.S5).” (Line 263-268).

A

B

Minor comments

3. Figure 2B, how was the serial dilution performed? If it was diluted into the equivalent amount of blood lysate the signal at low parasitemia becomes quenched more so (due to hemoglobin) than if the sample was diluted in PBS or equivalent (Azevedo 2014).

As described in the Methods section (Line 540-542), we performed a serial dilution with a

medium containing uninfected RBCs at the same hematocrit (2%) as that of the culture to normalize the background. In our case, as shown in Fig. 2B, the correlation was linear, at least between $2 \times 10^5 - 1 \times 10^2$ parasites and no effect on the signal was observed, even at low parasitemia.

4. A comment on the advantages and limitations of the gametocyte assay would be useful. For example, do you know if your assay is suitable for separating out early vs late stage gametocyte activity (e.g. Duffy et al, 2013, Malaria Journal).

Advantages:

The gametocytocidal assay with the GFP-NanoLuc line is a simple and user-friendly system that enables high-throughput screening with high sensitivity in any laboratory as long as standard equipment is prepared. Gametocyte purification with a magnet or percoll to remove the effect of background derived from uninfected RBCs and IFA to visualize parasites, both of which lower the throughput significantly, is not necessary for our assay system. In addition, expensive imaging devices to quantify parasite numbers, such as a high-content imaging system, are not necessary. Additionally, it is theoretically possible to investigate the antimalarial effect of compounds at the early gametocyte stage, which is achieved by shortening the culture time post-gametocyte induction, although we did not perform an assay with early-stage gametocytes in this study. In this case, the purification of stage I gametocytes may be required before seeding the parasites on an assay plate.

Limitations:

The only limitation of our protocol is that the costs for NanoLuc inhibitors provided by Promega are relatively high, which is not suitable for high-throughput screening with limited resources.

5. Figure 4D. It is very difficult to see the individual data points at the low end. A log scale, or a split axis would help this.

As suggested, we have revised Fig.4D and Fig.4E to clearly show the value at the low end (Line 773). New Fig.4D and 4E are shown below.

6. Figure 5C, the labels are positioned in such a way that they look like two data points in D7. As suggested, we have revised Fig.5C (Line 792-793). New Fig.5C is shown below.

Reviewer #3 (Remarks to the Author):

Miyazaki et al. present a versatile *Plasmodium falciparum* dual reporter line based on integrated bright Nanoluc luciferase and the fluorescent GFP (or mCherry). The line does not contain a selectable marker and shows reporter expression throughout the life cycle. The others nicely show that the constitutively expressed reporters do not appear to have any effect on the growth characteristics of the parasites at the different developmental stages of the life cycle. They propose that this line is suitable for the multi-stage evaluation of anti-malarial drug efficacy and using this line in HTS against asexual blood stage parasites and gametocytes, they identify OU0074008 as a novel anti-malarial compound.

While many different *P. falciparum* reporter lines have been developed, this new reporter line is attractive, because it is marker-less, has constitutive expression of GFP for visualization/isolation and the bright Nanoluc for bioluminescence read-out for drug assays at multiple stages of the parasite life cycle and appears to have a normal viability. This 'one-for-all' line could simplify screening of compounds directed against different parts of the life cycle of *P. falciparum*.

This manuscript represents a nice piece of work with investigations covering the full life cycle of the parasite. In my opinion however, some of the claims, especially claims regarding the performance of the parasite line during the liver stage part of the life cycle are not fully substantiated (see below).

We would like to thank the reviewer for the informative suggestions for improving our manuscript that we have incorporated in the updated manuscript.

Major comments

1. The analyses of the liver stage parasites by bioluminescence measurements appear to be limited. In the abstract (line 33) 'a quantitative' NanoLuc signal in liver stages (amongst other stages) is mentioned and it is stated that an assay system was established for asexual, gametocyte, and liver stages (line 35). In line 276 it is stated that luminescence and parasite number are well correlated, but this is not (clearly) shown for liver stages. And in the conclusion (lines 322 and 323) it is stated that a robust drug assay protocol is established for asexual-, gametocyte-, and liver-stage parasites. For asexual stages and gametocytes, relationship between parasite number and RLU, drug assay parameters (Z' factors etc.) have been described and multiple drugs were tested. However, for liver stage parasites, this is not the case. Only data of a single (?) drug experiment with atovaquone are shown. To substantiate the claim that a robust liver stage drug assay

based on bioluminescence with these parasites has been developed, more experimentation is needed. Similar to the work performed on the asexual- and gametocyte stages, the relationship between RLU values and parasite numbers needs to be assessed and multiple drug assays to determine activity with different drugs should be performed for liver stages as well.

In the revised manuscript we now include an analysis of parasite numbers versus luciferase signals at day 4, 5 and 6 post infection in Figure 5 (Line 792-793). In addition, we include assay parameters for the liver stage assay in Table 3 (Line 855). Lastly, we have tested PI4K inhibitor MMV390048 and compare its effect to those of atovaquone and a DMSO vehicle control. The new Figure 5 is shown below. We have also added new sentences related to the new Figure 5 in the Result section (Line 228-244).

2. Also, the claim that the activity of atovaquone against liver stage *P. falciparum* can be evaluated with a ‘higher sensitivity’ (line 301) compared to a previously established similar system by Marin-Mogollon et al (ref 14) needs more extensive experimentation (in fact IC50 values from both methods appear to be similar).

We agree with this comment and have revised the corresponding sentence accordingly. We did not validate that the liver-stage assay established in this study has a higher sensitivity when compared to a similar assay system that Marin-Mogollon et. al. previously established. As this sentence is misleading, we have rephrased it as follows (Line 330-334).

‘Moreover, we demonstrated that the activity of atovaquone against liver stage *P. falciparum* in primary human hepatocytes can be evaluated using our drug assay system with brighter luminescence when compared to a previously established system using Firefly luciferase.’

3. In the light of using a single reporter line for multiple stages of the life cycle, it would make sense to test the novel identified anti-malarial compound OU0074008 for asexual and gametocyte stages also in the liver stage assay.

We agree with this suggestion. In our initial trials, we tested OU0074008 with liver-stage parasites of another *P. falciparum* reporter line (not the GFP-NanoLuc line generated in this study) due to limitation in our laboratory, and we found that the compound has no anti-malarial effect on the liver-stage parasites. This data is included in Fig.S6 (Line 268-270, 957) and shown below. Thus, we considered that another compound known to be effective on liver-stage *P. falciparum* parasites is preferable for validating the liver-stage assay using the GFP-NanoLuc line than OU0074008. For this purpose, MMV390048, already known to have an anti-malarial effect on liver-stage *P. falciparum* (Paquet T et al. Sci Transl Med. 2017), was used for the additional experiment of liver-stage assay in this revision. MMV390048 showed a strong inhibitory effect on the GFP-NanoLuc line at the liver stage on the luminescence-based assay, which is comparable with the result of the conventional imaging assay. We added the new data about the new liver stage experiment to Fig.5D (Line 792-793, see above) and new sentences related to the new Figure 5 in the Result section (Line 228-244).

Minor points

1. The use of a single reporter line for the investigation of multiple stages of the life cycle would simplify the drug screening work. However, aren't the authors concerned that cross-contamination of stages (i.e. presence of remaining blood stages in the gametocyte preparation) could give problems in assessing drug activity against specific stages?

We believe that each assay established in this study showed stage-specific results. During gametocyte preparation, the asexual blood-stage parasites were killed by N-acetylglucosamine (NAG), as evidenced by the lack of antimalarial effects of atovaquone in the gametocyte-stage assay. If asexual stage parasites are left in the preparation, the luminescence intensity of the DMSO well should increase after 72 h of incubation, and there will be a difference in the signal intensity between the DMSO and the atovaquone wells.

2. For the figures with IC₅₀ curves, the following details should be clarified and mentioned: how many times was the assay performed (independently); how many wells were used per assay (96

or 384 well format); what is depicted, for example triplicates of wells or triplicates of assays?

We have revised this according to the suggestion by reviewer 2, to include biological and technical replicates in the Figure legends (Line 714, 732, 740-741, 757-758, 769, 788-789, 798-799, 801-802, 804-805, 809, 811, 836-838).

3. The authors mention also the development of a mCherry-NanoLuc line. They do show some characteristics of this line as well. While this is good to know, the line does not play any role in the main message of the manuscript. To keep the main text focused on the GFP line, the authors may consider to put all mCherry-NanoLuc data in one figure in the supplement and to just note that similar to the GFP-line, an mCherry line was developed with similar characteristics.

As suggested, we combined Fig.S1 and Fig.S3 and prepared a new Fig.S1 that describes the schematics of generation and reporter expression. New Fig.S1 is shown below.

4. The rationale for using NanoLuc is not completely clear from the introduction (also, please provide a ref for NanoLuc in line 89).

As per your suggestion, we have added new sentences in the introduction section (Line 88-91).

“NanoLuc is a suitable reporter protein for high throughput screening, which has a higher

luminescence intensity (80- to 240- fold) and structural stability than firefly luciferase (Hall, M.P., et al. (2012) ACS Chem Biol 7:1848).”

5. Line 112/113: was the *in vitro* growth of the mCherry-NanoLuc line also comparable to the WT strain?

As described in Fig.S1B, the mCherry-NanoLuc reporter line is a mixture of integrated parasites and NF54 WT. Hence, we did not examine the *in vitro* growth of the mCherry-NanoLuc line, although this line should show normal growth in the asexual blood stages.

6. Lines 166, 167: wouldn't these prolonged incubation times affect parasite viability? Did the authors check whether these gametocytes could still be transmitted? And what is the transmission rate of the gametocytes after the different incubation times?

We did not check the gametocyte viability and transmissibility after incubation with compounds because this is beyond the focus of the manuscript. We established a multi-stage drug assay including the asexual blood, gametocytes, and liver stages. However, since we are also interested in this point, we will consider it in the future.

7. Line 179: could the need for the NanoLuc inhibitor indicate prolonged stability of the protein? If so, do the authors think that it would be an option to develop a NanoLuc reporter with a degradation domain to counteract this issue?

Yes, we have considered using a NanoLuc reporter with a degradation domain (PEST) and generated a *P. falciparum* line expressing NanoLuc-PEST. However, contrary to our expectation, the NanoLuc protein derived from killed gametocytes accumulated due to the loss of proteasome degradation activity in the dead cells, thereby resulting in a higher luminescence signal in the positive control (epoxomicin) wells when compared to the DMSO wells. Therefore, we believe that the drug assay using gametocytes of the GFP-NanoLuc line with the extracellular NanoLuc inhibitor described in this manuscript is more robust and suitable for gametocytocidal assays.

8. Lines 225, 226: were all HSP70 parasites also GFP positive (indication of the stability of the transgenes in the genome)?

Double staining for GFP and HSP70 was not performed because the experiment was aimed to examine reporter expression. Since the GFP-NanoLuc line used in the liver stage experiment is clonal, all parasite population is likely to express reporter proteins.

9. Figure 5: panel E: on the y-axis % infected hepatocytes is stated. How was this calculated? What is the number of liver stages per well? Is this a single assay from which means and SD of

triplicate wells are shown? I would strongly suggest using a more commonly used parameter, such as % of untreated control instead.

In light of this comment and the comments raised by reviewer 2, we have adapted Figure 5G to show the absolute numbers of parasites (See above). In addition, we have included a panel 5F that shows the effect of atovaquone on the total parasite area (See above). These experiments show data from three independent replicates, and we have clarified this in the Figure legend.

10. Lines 299-302: the authors state that the liver stages could be measured with high sensitivity. I do think that more data are needed for this statement (see above). Also, the RLU values appear to be low (around 4000) if these reflect parasite numbers around 50-100 per well. In papers describing Nanoluc in liver stage parasites from other malaria parasites (de Niz et al, Malar J. 2016;15:232 and Voorberg-van der Wel et al, Anal Chem 202, 92, 9, 6667-6675) RLU values are much higher. Can the authors comment? How long did the authors measure per well? Did the authors deviate from the protocol of the manufacturer (and/or the other papers)?

As we replied to the comment raised in 'major point 2,' since this sentence is misleading, we have rephrased it as follows (Line 330-334):

“Moreover, we demonstrated that the activity of atovaquone against liver stage *P. falciparum* in primary human hepatocytes can be evaluated using our drug assay system with brighter luminescence when compared to a previously established system using Firefly luciferase.”

Regarding the RLU value, thank you for pointing it out. Absolute values of RLU are incomparable between our study and other studies because it varies depending on the detection system and measurement settings (dynamic range) of luminometers used. In addition, since the promoter and the *Plasmodium* species used in each study differ from those of our GFP-NanoLuc line, it becomes even more challenging to compare the RLU values. Even if the signal intensity is high, it may be because the background signal is high. Relative values of RLU (i.e. Signal/Background: S/B) are more important when comparing the assay quality. We consider that the liver-stage assay using the GFP-NanoLuc line has very high quality since the S/B is high enough (Table 3).

11. Figure 2, panel B: the RLU values of low numbers of iRBC are below the background value of uninfected RBC. Do the authors have any idea why this is the case?

Thank you for spotting this. We carefully checked our original data and noticed that the background signal in the asexual blood stage was much lower than 10^3 . We revised Figure 2B according to the original data. New Fig.2B is shown below.

12. M&M, lines 394, 395: can the authors give a few more details here? Was this a mixed population of parasites, or for example, purified schizonts? What was the starting parasitemia (and much time did it take until parasitemia reached 5%)?

To address this comment, we have now rephrased the methods section as follows (Line 427-429): “Subsequently, plasmid-loaded RBCs were incubated with mixed stages of *P. falciparum*-infected RBCs with 0.5% parasitemia and cultured *in vitro* for three days.”

13. M&M, lines 487-490: why did the authors use Matrigel one day after hepatocyte seeding and after parasite inoculation? Didn't the removal of Matrigel affect the hepatocyte monolayer?

Matrigel is added after the hepatocyte cell layer is formed, removed before infection to allow for better sporozoite-hepatocyte contact and added again after infection to maintain the quality of the infected cultures. With the selected batches of primary human hepatocytes that we use, the removal of Matrigel does not affect the hepatocyte monolayer.

14. M&M, lines 497: was the counting performed manually or did the authors use a script?

Liver stage parasite counting was performed manually. We have now added this information to the method section (Line 532-535)

“Parasite size was determined manually as parasite area in μm^2 using a Cell Insight High Content Screening platform equipped with the Studio HCS software (Thermo Fisher Scientific) in Celis Platform (ICM, La Pitié-Salpêtrière, Paris) as described previously.”

15. M&M, lines 518-520: what was the end parasitemia of the assay?

When the NanoLuc assay was performed, we did not measure absolute parasitemia in the 96- or 384-well plates. In this assay, all the reaction samples were used for the detection of luminescence.

REVIEWERS' COMMENTS:

Reviewer #1 (Remarks to the Author):

The authors have addressed all the issues that I previously raised. I have only two further minor comments:

Figure 5C – Figure legend indicates this as being the “Area of NF54 WT and GFP-NanoLuc exoerythrocytic form (EEF) parasites at different time points of development within primary human hepatocytes” but in the YY axis it reads “length (μM)”. Please correct this.

Figure 5F – The way that “total area (μm^2)” was calculated should be defined in the figure legend.

Reviewer #2 (Remarks to the Author):

The authors have provided suitable responses to the majority of my initial questions. However, one of the responses needs to be included in the manuscript, and one of the responses needs minor clarification. See specific comments below.

Regarding this initial question "A comment on the advantages and limitations of the gametocyte assay would be useful. For example, do you know if your assay is suitable for separating out early vs late stage gametocyte activity (e.g. Duffy et al, 2013, Malaria Journal)." The answers provided are acceptable, however, the intention was for the authors to include a similar answer in the discussion.

Regarding the number of replicates in each figure. I would like to clarify that it currently appears like each biological replicate generally consisted of a single sample (e.g. a single well), and that this was performed several times independently. While it is acceptable to use a single well in each experiment if there is suitable number of biological replicates, it is unusual not to use 2-3 technical replicates in each experiment since it reduces variation in the system, hence I wanted to clarify this is how the authors performed the experiments.

Reviewer #3 (Remarks to the Author):

The authors have adequately addressed the points raised for the first manuscript.

Just a single remark: please correct the label of the y-axis in fig 5C (according to what is mentioned in lines 532/533 of the manuscript).

Response letter for the manuscript entitled "A versatile *Plasmodium falciparum* reporter line expressing NanoLuc enables highly sensitive multi-stage drug assays"

Our responses are indicated in a blue font.

Reviewers' comments:

We sincerely thank the Reviewers and Editors for their comments and suggestions on our manuscript. In response to the issues raised by the reviewers, we revised the manuscript to address the comments raised. We believe that the latest manuscript will be suitable for publication.

Reviewer #1 (Remarks to the Author):

The authors have addressed all the issues that I previously raised. I have only two further minor comments:

Figure 5C – Figure legend indicates this as being the “Area of NF54 WT and GFP-NanoLuc exoerythrocytic form (EEF) parasites at different time points of development within primary human hepatocytes” but in the YY axis it reads “length (μM)”. Please correct this.

We confirmed the y-axis indicates the length of the liver stage parasites. We have revised the related sentences according to this information (Line 815-820).

Figure 5F – The way that “total area (μm^2)” was calculated should be defined in the figure legend. We have revised the Figure legend to clarify the way to calculate the total area (Line 826-829).

Reviewer #2 (Remarks to the Author):

The authors have provided suitable responses to the majority of my initial questions. However, one of the responses needs to be included in the manuscript, and one of the responses needs minor clarification. See specific comments below.

Regarding this initial question "A comment on the advantages and limitations of the gametocyte assay would be useful. For example, do you know if your assay is suitable for separating out early vs late stage gametocyte activity (e.g. Duffy et al, 2013, Malaria Journal)." The answers provided are acceptable, however, the intention was for the authors to include a similar answer in the discussion.

Thank you for your comment. We included our answer regarding the advantages and limitations of the gametocyte assay in the discussion section (Line 324-338).

Regarding the number of replicates in each figure. I would like to clarify that it currently appears like each biological replicate generally consisted of a single sample (e.g. a single well), and that this was performed several times independently. While it is acceptable to use a single well in each experiment if there is suitable number of biological replicates, it is unusual not to use 2-3 technical replicates in each experiment since it reduces variation in the system, hence I wanted to clarify this is how the authors performed the experiments.

We have confirmed the definition of the biological and technical replicates and thoroughly updated the corresponding part in the manuscript.

Reviewer #3 (Remarks to the Author):

The authors have adequately addressed the points raised for the first manuscript.

Just a single remark: please correct the label of the y-axis in fig 5C (according to what is mentioned in lines 532/533 of the manuscript).

We confirmed the y-axis indicates the length of the liver stage parasites. We have revised the related sentences according to this information (Line 815-820).